# Convolution Goes Higher-Order: A Biologically Inspired Mechanism Empowers Image Classification

**Simone Azeglio**
Institut de la Vision & Laboratoire des Systèmes Perceptifs
INSERM, CNRS, Sorbonne University & École Normale Supérieure - PSL
17 Rue Moreau, Paris 75012 & 29 Rue d'Ulm Paris 75005
simone.azeglio@gmail.com

**Olivier Marre**
Institut de la Vision
INSERM, CNRS, Sorbonne University
17 Rue Moreau, Paris 75012

**Peter Neri**
CHT Erzelli & Laboratoire des Systèmes Perceptifs
IIT & École Normale Supérieure - PSL
Via Enrico Melen 83, Genova 16152 & 29 Rue d'Ulm Paris 75005

**Ulisse Ferrari**
Institut de la Vision
INSERM, CNRS, Sorbonne University
17 Rue Moreau, Paris 75012

## Abstract

We propose a novel enhancement to Convolutional Neural Networks (CNNs) by incorporating learnable higher-order convolutions inspired by nonlinear biological visual processing. Our model extends the classical convolution operator using a Volterra-like expansion to capture multiplicative interactions observed in biological vision. Through extensive evaluation on standard benchmarks and synthetic datasets, we demonstrate that our architecture consistently outperforms traditional CNN baselines, achieving optimal performance with $3^{rd}/4^{th}$ order expansions. Systematic perturbation analysis and Representational Similarity Analysis reveal that different orders of convolution process distinct aspects of visual information, aligning with the statistical properties of natural images. This biologically-inspired approach offers both improved performance and deeper insights into visual information processing.

## 1 Introduction

*"The correlation over space translates into a correlation over order."* Koenderink and van Doorn [1]

Convolutional Neural Networks (CNNs) have become the cornerstone of modern computer vision tasks, demonstrating remarkable performance across a wide range of applications [2], [3], [4]. The success of CNNs is largely attributed to their ability to capture local patterns and hierarchical features in images through their layered structure and weight sharing mechanism [5], [6].

39th Conference on Neural Information Processing Systems (NeurIPS 2025).

However, natural images contain complex correlations and higher-order statistics that extend beyond simple linear relationships [7], [8], [9]. These include texture patterns, edge interactions, and intricate geometric structures that are prevalent in real-world visual scenes. Importantly, these correlations exhibit a hierarchical structure, with higher-order correlations becoming increasingly sparse [1]. Traditional CNNs, particularly those with limited depth, often struggle to effectively exploit these higher-order correlations [10], [11], [12]. The limitation of standard CNNs in capturing higher-order correlations stems from their reliance on linear convolutions followed by pointwise nonlinearities [13]. Although deeper networks can approximate complex functions through the composition of many layers, this approach may be computationally expensive and require large amounts of training data. To address this limitation, we propose a learnable extension of the classical linear convolution operator to incorporate higher-order terms, akin to a Volterra expansion [14], [15]. This approach allows for the direct modeling of multiplicative interactions between neighboring pixels, enabling the network to capture complex local structures more effectively, even in shallow architectures.

We design models by incorporating higher-order convolutions, and compare their performance against standard CNN baselines. We evaluated our models on a range of datasets, beginning with synthetic images typically used to assess sensitivity to higher-order correlations. Subsequently, we extended our experiments to compare our model against CNN baselines on widely-used datasets including MNIST, FashionMNIST, CIFAR10, CIFAR100, Imagenette and Imagenet. Our findings reveal that these higher-order convolutions improve image classification performance, offering a promising direction for advancing deep learning models in computer vision tasks.

Interestingly, our results show optimal performance at the $3^{\text{rd}}$ or $4^{\text{th}}$ order, aligning remarkably with the distribution of pixel intensities in natural images as analyzed by Koenderink and van Doorn [1] and Tkačik et al. [16]. This alignment suggests that our approach effectively captures the fundamental structure of visual information in the natural world. Through systematic perturbation analysis, we isolate the contributions of specific image statistics to model performance, providing insight into how different orders of convolution process distinct aspects of visual information. Furthermore, we employ Representational Similarity Analysis (RSA) [17] to investigate the internal representations learned by our model compared with standard CNNs, revealing distinct geometries that indicate different strategies for encoding and processing visual information.

Our approach addresses the limitations of simple pointwise nonlinearities in capturing local image structure, especially in networks of limited depth or width. In particular, we show that incorporating such computations at the earliest stages of image processing, even within the first layer of a neural network, yields significant benefits. This finding suggests that models benefit from embedding these computations via direct engagement with the *superficial structure* of images [18].

## 1.1 Biological Inspiration and Related Work

Our work draws inspiration from biological visual systems, where processing extends beyond simple pointwise nonlinearities and first-order convolutions [19, 20, 21, 22, 23]. These non-pointwise nonlinearities enable more effective extraction and integration of spatial information [24, 25, 26, 27], beginning in the retina [21] and continuing in the visual cortex where specialized neurons respond to complex features like corners or line endings [28].

In computational research, the concept of nonlinear receptive fields has been explored through various approaches. Berkes and Wiskott [29] analyzed inhomogeneous quadratic forms as receptive fields, while Li et al. [30] provided theoretical insights on Volterra-based convolutions in neural networks. Zoumpourlis et al. [31] implemented Volterra-inspired convolutions primarily in the first layer of a Wide ResNet-28, but failed to demonstrate significant performance improvements and limited their analysis to second-order expansions. In contrast, our work extends this foundation by implementing and systematically analyzing expansions up to the $4^{\text{th}}$ order, where we demonstrate that $3^{\text{rd}}/4^{\text{th}}$ order expansions yield optimal performance—a finding we uniquely connect to the statistical properties of natural images documented by Koenderink and van Doorn [1].

For video classification, Roheda et al. [32] introduced Volterra Neural Networks using cascaded $2^{\text{nd}}$-order filters with low-rank approximations for spatiotemporal tasks, creating separate spatial and temporal streams that are later fused. Our approach differs fundamentally, as we focus on static image classification with direct implementation of higher-order terms (up to $4^{\text{th}}$ order) within the convolution operation itself, preserving parameter independence across orders for more flexible feature learning.

While Roheda et al. [32] emphasize computational efficiency through approximations, our work provides deeper insights through systematic perturbation analysis and representational similarity analysis that reveals how different orders process distinct aspects of visual information. Related developments in fine-grained image classification include bilinear CNN models [33] and compact bilinear pooling [34], with applications extending to computational neuroscience [35].

Our approach extends and complements these previous higher-order methods by embedding higher-order operations directly within the convolutional operator, creating independent feature maps for each order that are combined before nonlinearities. This design choice enables the network to learn and apply complex nonlinear transformations that better constitute an *algorithmic implementation* of biological visual processing. Through our structured perturbation analysis, representation geometry analysis, and controlled texture experiments, we provide novel insights that go beyond performance improvements, demonstrating *why* and *how* higher-order processing benefits visual representation.

## 1.2 Beyond Pointwise Nonlinearities

Traditional convolutional neural networks typically employ pointwise nonlinearities with weighted sums of inputs as arguments. Expanding these nonlinearities as a polynomial series reveals that different orders of the expansion share the same weights, effectively coupling the terms across orders and leading to inter- and extra-order dependencies:

$$
\sigma \left( \sum_{i=0}^{2} w_i x_i \right) \approx \alpha_0 + \alpha_1 (w_1 x_1 + w_2 x_2) + \alpha_2 (w_1 x_1 + w_2 x_2)^2 + \ldots
$$
$$
= \alpha_0 + \alpha_1 w_1 x_1 + \alpha_1 w_2 x_2 + \quad \alpha_2 w_1^2 x_1^2 + \alpha_2 w_2^2 x_2^2 + 2\alpha_2 w_1 w_2 x_1 x_2 + \ldots
$$
(1)

This *tied-weight* issue becomes particularly problematic when the network is not *deep enough* or *wide enough* to compensate, as the precise bounds for *enough* depth or width depend on the universal approximation theorem and are often challenging to determine exactly [36]. Consequently, the standard pointwise nonlinearity with linear summation may fail to capture complex relationships in the data, especially in shallower or narrower networks. To address this limitation, we propose a novel approach that can be viewed as an implementation of non-pointwise nonlinearities. Our method, which will be described in detail in **Section** 2, can be seen as a *learnable* generalized 2-D detector model [25] representing a receptive field that responds to multiplicative interactions between inputs, similar to how some biological neurons process visual information (see **Subsection** 1.1). This approach offers a solution for capturing complex correlations at the level of a single mechanism, allowing for more flexible and powerful representations even in compact network architectures.

## 2 Higher-Order Convolution

To perform more complex computations while preserving the benefits of locality and weight sharing that are inherent in standard convolutions, we propose the concept of higher-order convolution. This approach extends the traditional convolution operator to include higher-order terms, allowing for more sophisticated feature extraction.

In order to explain our higher-order convolutional layer, we start by considering an input image patch **P** with $n$ elements, reshaped as a vector **x**. Standard convolution can be expressed in terms of linear filtering, relating input **x** and output **y** as follows: $y(\mathbf{x}) = b + \sum_{i=1}^{n} w_1^i x_i$ where $w_1^i, i = 1, ..., n$ represents the weights of the convolution kernel. We expand this function to include quadratic, cubic, and higher-order terms. The general form of this expansion is:

$$
y(\mathbf{x}) = b + \sum_{i=1}^{n} w_1^i x_i + \sum_{j=1}^{n} \sum_{i=1}^{n} w_2^{ij} x_i x_j + \sum_{k=1}^{n} \sum_{j=1}^{n} \sum_{i=1}^{n} w_3^{ijk} x_i x_j x_k + ...
$$

Extending this to the entire image, we can express the higher-order convolution operation as:

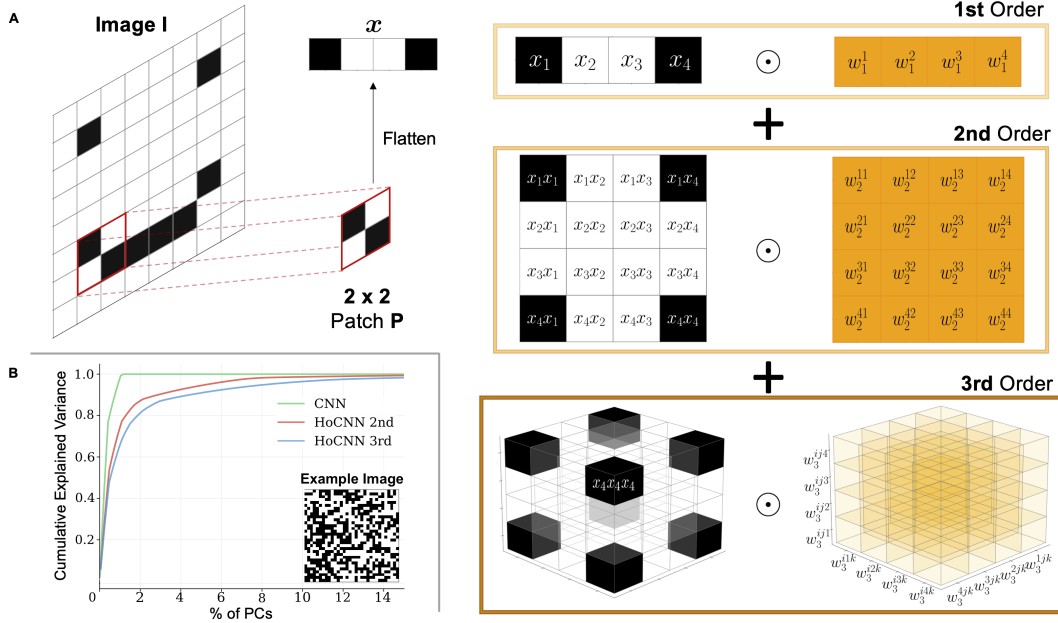

Figure 1: **Extending classical convolution** (A) Implementation of higher-order convolution: Input patch is flattened to vector **x**. 1$^{st}$ order uses classical convolution with weight-sharing. 2$^{nd}$ and 3$^{rd}$ orders compute outer products - respectively one and two times - of **x** with itself and dot product with the corresponding weights. Feature maps are then summed before ReLU activation. (B) Cumulative explained variance of principal components for standard CNN model vs Higher-order CNN models with 2nd and 3rd expansion quantitatively confirms the tied-weight issue for classical models. Inset: Example of the 32×32 binary texture image used in this analysis, containing all possible 1-, 2-, 3-, and 4-point correlations for 2×2 patches.

$$Y_{l,m} = b + \sum_{i,j} w_1^{ij} X_{l+i,m+j} + \sum_{i,j} \sum_{k,h} w_2^{ijkh} X_{l+i,m+j} X_{l+k,m+h}$$
$$+ \sum_{i,j} \sum_{k,h} \sum_{p,q} w_3^{ijkhpq} X_{l+i,m+j} X_{l+k,m+h} X_{l+p,m+q} + \dots \quad (2)$$

Here, $Y_{l,m}$ represents the output at position $(l,m)$ in the feature map, $X$ is the input image, and $w_1$, $w_2$, $w_3$ are the learnable weights for the first, second, and third-order terms respectively. The sums over $i, j, k, h, p, q$ run over the dimensions of the convolution kernels for each order.

By symmetry considerations, the total number of parameters in this expansion grows as: $n_V = \frac{(n+p)!}{n!p!}$ where $p$ represents the order of the term and $n$ the size of the kernel. To address potential issues with the relative magnitudes of higher-order terms, we implement a normalization strategy. For kernels of order greater than the 1$^{st}$, we apply a scaling factor $s$ calculated as $s = \frac{1}{\sqrt{n_V}}$. The scaling factor helps to balance the contribution of higher-order terms relative to the first-order (linear) terms, promoting stability and preventing higher-order terms from dominating network behavior during training. **Figure** 1 **A** illustrates this concept for binary image, showing how the classical convolution is extended to include higher-order terms for a single patch. In practice, this operation is applied across the entire image, resulting in a feature map for each order of the expansion, which are then summed together before a standard pointwise nonlinearity as ReLU (see for example, **Figure** 2 **B**). This layer can functionally replace a standard convolutional layer in neural network architectures and is compatible with training through backpropagation.

## 2.1 A Quantitative Perspective on the Tied-Weight Issue

To demonstrate how a Higher-order CNN (HoCNN) addresses the tied-weight issue introduced in **Section** 1.2, we conducted a systematic analysis of weight independence in the network's representations. We generated a fixed input image ($32 \times 32$ pixels) containing binary textures exhibiting all possible 1-, 2-, 3-, and 4-point correlations for 2×2 patches (see inset in Figure 1B), corresponding to fixing the $x_i$ terms in Equation 1. We then randomly initialized the same model architecture 10,000 times and analyzed the activations after the convolutional block (Conv/HoConv + BatchNorm + ReLU + Max Pooling) for two architectures: a CNN with 10 kernels of size 2×2 and a HoCNN with 2 kernels of size 2×2 with higher-order expansions.

To quantify the degree of weight independence, we employed Principal Component Analysis of these activations, measuring the number of principal components (PCs) needed to explain 95% of the variance. Our hypothesis was that tied weights would reduce the number of principal components, while greater weight independence would require more components to capture the same variance. The results confirmed our expectations (see **Figure** 1 **B**): the CNN required only **0.9%** of PCs (87 out of 9610) to explain 95% of the variance, while the HoCNN required **5.3%** of PCs (102 out of 1922) with second-order expansion and **8.3%** (159 out of 1922) with third-order expansion. While the CNN has more total components due to its higher channel count (5×), additional analyses with matched channel counts yielded similar results. These findings quantitatively demonstrate that higher-order convolutions introduce more independent weights, leading to a richer representation space with reduced weight coupling. The increasing number of components needed for variance explanation directly shows how our approach mitigates the tied-weight issue inherent in standard CNNs. Additional analyses with various nonlinearities showed consistent results, further supporting these conclusions (see **Appendix** A.11).

## 3 Experiments & Results

We evaluated our model on a synthetic texture dataset with higher-order correlations and multiple standard benchmarks: MNIST, FashionMNIST, CIFAR-10, CIFAR-100, and ImageNet. Additional results on Imagenette and fine-grained classification (CUB-200-2011) are provided in **Appendix** A.6 and A.7.

### 3.1 Synthetic Dataset: Structured Visual Textures

To test our model's sensitivity to higher-order correlations, we generated a synthetic dataset of structured visual textures based on the work of Victor and Conte [37] (see **Figure** 2). These textures allow precise control over the type and intensity of contained correlations. To generate the textures, we utilized a custom software library [38] implementing the method developed by Victor and Conte [37]. This process involves sampling from a distribution of binary textures with specific multipoint correlation probabilities while maximizing entropy. The correlation intensity is parameterized by parity counts of white or black pixels within 1-, 2-, 3-, or 4-pixel tiles (termed *gliders*, **Figure** 2)). Notably, two-point and three-point gliders create distinct correlations through different spatial configurations: horizontal, vertical, or oblique for two-point, and L patterns with various orientations for three-point correlations. Our synthetic dataset comprises 2000 training images, 1000 validation images, and 2000 testing images. All reported results are based on the test set.

We framed the problem as an image classification task with 10 different classes corresponding to various N-point correlations (N ranging from 1 to 4). To establish a baseline, we first tested a CNN model consisting of 3 blocks (2 convolutional - 1st conv layer with 10, 2x2 kernels and 2nd conv layer with 2, 8x2 kernels; both followed by batch normalization, ReLu nonlinearity and max pooling and 1 fully connected layer). Our results indicate (see **Table** 1 and **Figure** 2) that this basic architecture fails to correctly discriminate among the different 2-point and 3-point correlations. We then repeated the classification task using three different HoCNNs with expansions up to the 2nd , the 3rd , and the 4th order kernels (2 kernels of size 2x2) and an additional convolutional (conv layer with 2 kernels of size 8x2 + batch norm. + ReLU + pooling) and fully connected layer replicating the structure of the baseline CNN. These networks produced increasingly good performance (**Table** 1 and **Figure** 2), highlighting the need for our proposed higher-order convolution approach. For completeness, we report the number of parameters of the different architectures in **Table** 1.

Table 1: CNN and HoCNN performances on Texture classification

| Model | Accuracy (%) | # Params |
|---|---|---|
| CNN (baseline) | 59.14 | 492 |
| HoCNN (up to 2nd order) | 82.42 | 293 |
| HoCNN (up to 3rd order) | 89.02 | 488 |
| HoCNN (up to 4th order) | 92.32 | 1259 |

Figure 2: **Multipoint correlations and glider classification** (A) Textures generated with N-point gliders (N ranging from 1 to 4), totaling 10 classes when taking into account parity constraints. (B) Confusion matrices for different models, from top left: baseline CNN; top right: Higher-order CNN (HoCNN) with kernels expanded up to the 2nd order; bottom-left: HoCNN with kernels expanded up to the 3rd order; bottom-right: HoCNN with kernels expanded up to the 4th order. Taken together the four confusion matrices show that higher-orders progressively allow our network to properly capture relevant features for image classification.

## 3.2 Benchmark Datasets

**MNIST, FashionMNIST, CIFAR-10, CIFAR-100**
To further validate our approach and ensure its generalizability, we conducted extensive experiments on four well-established benchmark datasets: MNIST, FashionMNIST, CIFAR-10, and CIFAR-100. We focused on these relatively small datasets to allow for comprehensive testing and analysis. To ensure the robustness of our results and account for the inherent variability in neural network training, we conducted multiple independent runs for each experiment. Specifically, we initialized and trained each model with 50 different random seeds. This approach provides a more reliable estimate of model performance by mitigating the effects of random initializations.

We compared our HoCNN with 3rd order kernels (3×3) against a standard CNN baseline with equivalent kernel size. Notably, extending to 4th order kernels showed no significant improvement in test accuracy. Complete architectural and training details are provided in **Appendix** A.9, with parameter comparisons in **Table** 2. Test accuracy results, averaged over 50 realizations (**Figure** 3 and **Table** 2), demonstrate consistent performance advantages for HoCNN across all datasets, with particularly notable improvements on the more complex CIFAR-10 and CIFAR-100 tasks. These results suggest fundamentally different representational capabilities in HoCNN (further explored in **Section** 4). Additional experiments with a deeper and more complex CNN architecture, detailed in **Appendix** A.2, showed inferior performance compared to our simpler HoCNN, highlighting the efficiency of higher-order feature extraction.

*Training Details.* For all experiments, we used AdamW optimizer [39] with learning rate 0.001, weight decay 5e-4, batch size 64, cross-entropy loss and early stopping. Images were normalized using z-score standardization without data augmentation. Experiments used an NVIDIA RTX 4080

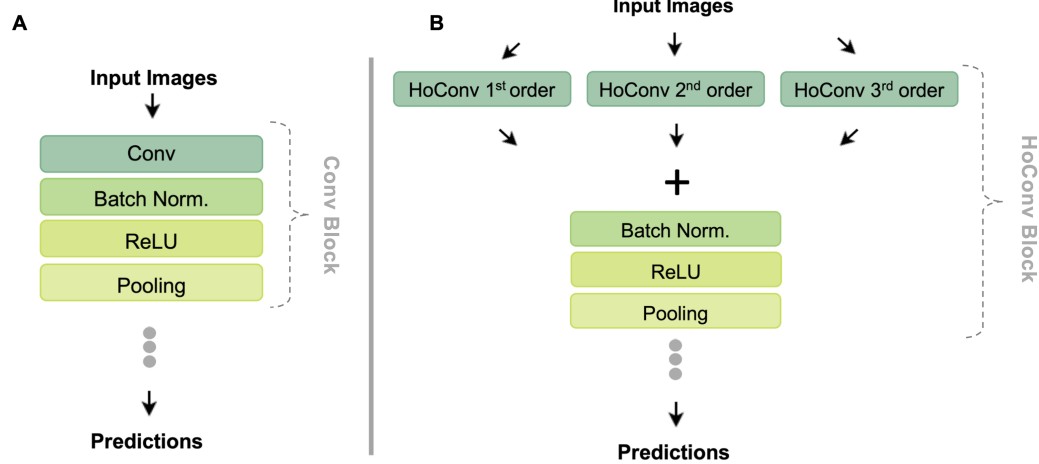

Figure 3: (A) Comparison of standard convolution (Conv) and (B) higher-order convolution (HoConv) blocks.

Table 2: Accuracy (with $\sigma$, standard deviation over multiple initializations) of CNN and HoCNN models across image classification benchmarks. MNIST, FashionMNIST, CIFAR-10, CIFAR-100: 50 runs; ImageNet: 5 runs.

| Dataset | Accuracy (%) $\pm \sigma$ | | # Params | |
|---|---|---|---|---|
| | *CNN* | *HoCNN* | *CNN* | *HoCNN* |
| MNIST | $99.13 \pm 0.09$ | $99.30 \pm 0.07$ | 82,706 | 80,262 |
| FashionMNIST | $90.48 \pm 0.32$ | $90.95 \pm 0.35$ | 82,706 | 80,262 |
| CIFAR-10 | $69.76 \pm 0.64$ | $72.87 \pm 0.54$ | 82,706 | 80,262 |
| CIFAR-100 | $36.92 \pm 0.71$ | $40.42 \pm 0.84$ | 385,006 | 383,478 |
| Imagenet | $68.62 \pm 0.08$ | $69.47 \pm 0.09$ | 11,689,512 | 11,676,252 |

Ti GPU, with training times of 10-15 minutes for MNIST/FashionMNIST and 20-30 minutes for CIFAR-10/CIFAR-100 (HoCNN requiring 1.5-2× longer). Complexity analysis is in **Appendix** A.2.

**IMAGENET**
To investigate the scalability and real-world applicability of our approach, we implemented Higher-order Convolution in a ResNet-18 [4] architecture (HoResNet-18) and evaluated it on ImageNet [40]. To ensure statistical robustness, we conducted 5 independent training runs with different random seeds. HoResNet-18 achieved $69.47\% \pm 0.09\%$ test accuracy compared to ResNet-18's $68.62\% \pm 0.08\%$, demonstrating a statistically significant improvement of 0.85% (see **Table** 2). Results on Imagenette and fine-grained classification (CUB-200-2011) are reported in **Appendix** A.6 and A.7.

HoResNet-18 maintains the standard ResNet-18 structure, but uses higher-order residual blocks in its first stage and standard blocks thereafter. This hybrid design preserves ResNet's architecture while incorporating higher-order convolutions. Parameter counts are shown in **Table** 2, with architectural details in **Appendix** A.9.1. These results provide strong evidence for the efficacy of our higher-order convolution approach, demonstrating successful scaling to deeper architectures and large-scale datasets. To further investigate scalability, we conducted extensive experiments with deeper ResNet architectures (ResNet-34 and ResNet-50), confirming that performance improvements persist across model depths (see detailed results in **Appendix** A.4).

*Training Details.* For ImageNet, we used standard practices: normalization with mean = [0.485, 0.456, 0.406] and standard deviation = [0.229, 0.224, 0.225]; Stochastic Gradient Descent with momentum = 0.9 as optimizer. We conducted 5 independent runs with different random seeds to ensure statistical robustness. The ImageNet experiments were run on an NVIDIA A100 GPU, requiring approximately 2-3 days per run. The detailed analysis of computational requirements and FLOP measurements for different orders of convolution can be found in **Appendix** A.2.

# 4 Image Statistics Sensitivity and Neural Representations

To understand how our Higher-order Convolutional layer (HoConv) processes information differently from standard Convolutional (Conv) layers, we conducted two complementary analyses: an investigation of sensitivity to image statistics through perturbations (akin to adversarial attacks) and a study of representational geometry.

HoCNN's sensitivity to higher-order image statistics has been quantified by conducting a systematic perturbation analysis on the CIFAR-10 test dataset using both architectures (50 pretrained models each), while keeping the network trained on the unperturbed images. We generated synthetic textures with varying orders of statistical correlations (1-, 2-, 3-, and 4-point) and interpolated them with test images at different intensity (I) levels ($\text{img}_{perturbed}$ = (1-I)×$\text{img}_{CIFAR-10}$ + I×$\text{img}_{perturbation}$ , with I $\in [0.05, 0.20]$). While HoCNN achieves superior performance on CIFAR-10 test set images (see **Table** 2), it shows increased vulnerability to higher-order statistical perturbations. The performance degradation becomes progressively more pronounced as we move from lower to higher-order correlations, with the gap widening at higher perturbation intensities (see **Figure** 4 **A** and **B** for I = 0.12). For instance, with third-order perturbations at I = 0.12, HoCNN's accuracy drops by 78.0% compared to CNN's 73.7% (complete results in **Appendix** A.12). In Appendix A.5, we complement this analysis by showing how HoCNNs are tipically more robust than CNNs for standard corruptions in CIFAR-10C and CIFAR-100C [41]. This enhanced sensitivity to statistical perturbations suggests HoCNN's increased capacity to leverage higher-order patterns. To better understand how this statistical sensitivity translates into improved classification performance, we examined the model's internal representations using Representational Similarity Analysis (RSA) [17] on CIFAR-10, analyzing activations from 100 test images (10 per class) averaged across 50 model realizations. The Conv and HoConv blocks exhibit distinct representational geometries (**Figure** 4A & B), with the latter showing more pronounced class-specific patterns. Analysis of pairwise distance distributions (**Figure** 4D) reveals that HoConv representations are significantly more dispersed in the high-dimensional space, indicating its ability to capture more diverse and discriminative features. Detailed analysis of individual order components and layer-wise correlations is presented in **Appendix** A.13.

Together, these analyses demonstrate that HoCNN's superior classification performance stems from its ability to capture and leverage higher-order statistical features in natural images, resulting in richer and more discriminative internal representations.

# 5 Discussion and Limitations

Our study introduces a novel approach to image classification by extending convolutional neural networks with higher-order learnable convolutions, inspired by complex nonlinear biological visual processing. Our findings indicate that expansion beyond the $4^{th}$ order is unnecessary, a conclusion supported by both theoretical insights and empirical evidence. Koenderink and van Doorn [1]'s analysis of natural image statistics reveals that the quadratic, cubic and quartic power dominates approximately 63%, 35%, and 2% of the pixels, respectively. Our experimental results align with this distribution, showing no significant performance gains beyond the $3^{rd}$ order in benchmark tasks and the $4^{th}$ in structured visual textures. Through systematic perturbation analysis, we validate this alignment by isolating the contributions of specific image statistics to model performance.

Our approach to structured visual textures connects to mechanistic models in computational neuroscience, particularly those studying the Drosophila visual system's sensitivity to 3-point correlations [19, 20]. This connection strengthens the relevance of our model and highlights its potential to provide insight into visual processing in both biological and artificial systems. The consistent outperformance of our HoCNN across various datasets demonstrates the effectiveness of incorporating these biologically inspired computations. RSA reveals distinct geometries, providing evidence that our model processes visual information differently from standard CNNs.

## 5.1 Limitations

Despite the demonstrated benefits, our higher-order convolution approach has several limitations worth acknowledging. The primary challenge is computational complexity, as detailed in Appendix A.2. Our FLOP analysis reveals that $2^{nd}$ and $3^{rd}$ order operations require approximately 5× and 18× more computation than standard convolutions, respectively. While we mitigate this by strategic

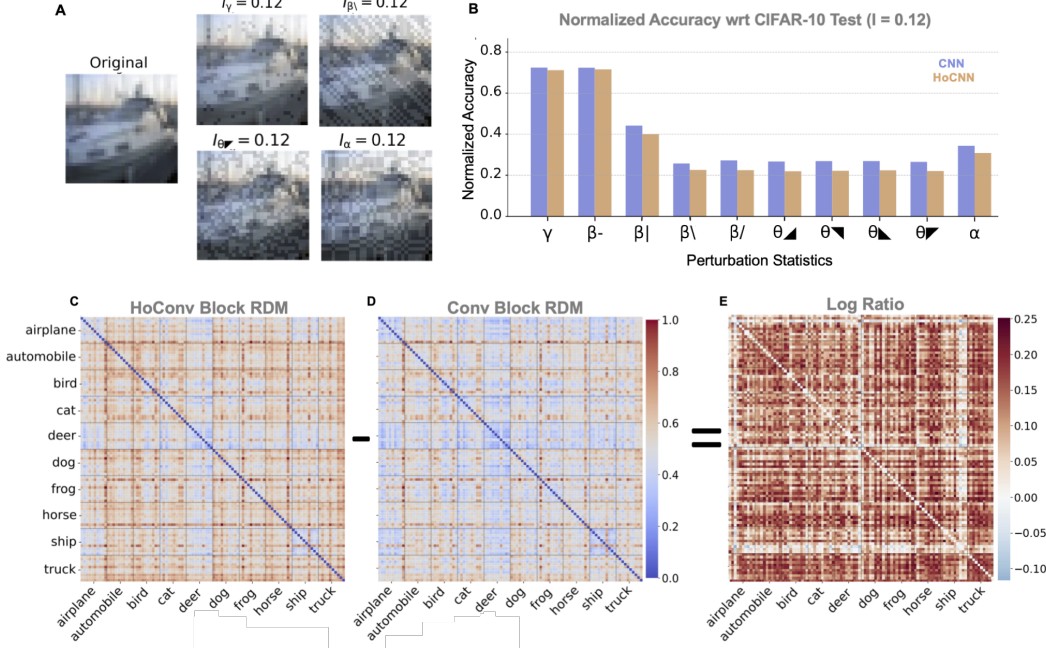

Figure 4: **Perturbation Analysis and Neural Representations**. (A) An exemplar test image (CIFAR-10) and perturbed examples with different 1, 2, 3, 4-point correlations statistics and common intensity (I = 0.12). (B) Normalized (wrt unperturbed CIFAR-10) accuracy for Convolutional (Blue) vs Higher-order (Orange) networks: performances of Higher-order network are systematically worse for more structured perturbations at fixed intensity (I = 0.12). (C) Representational Dissimilarity Matrix (RDM) for baseline Convolutional Block (Conv layer, Batch Norm, ReLU, Pooling). (D) RDM for the Higher-order Convolutional block. (E) Log Ratio between the two RDMs, capturing different representational geometries between the two blocks.

placement of higher-order operations in early network layers, this computational overhead represents a significant trade-off for resource-constrained applications. The increasing parameter count with higher-order expansions also introduces scalability challenges. Although our implementation leverages kernel symmetry to reduce parameters (e.g., reducing the 3$^{\text{rd}}$ order parameter count by 4.42×), memory requirements still increase substantially with order. This limits practical applications beyond 4$^{\text{th}}$ order expansions, though our experiments indicate minimal benefits beyond this point.

Additionally, our approach shows increased vulnerability to higher-order statistical perturbations (Section 4), suggesting a trade-off between enhanced feature extraction capability and certain forms of robustness. We also observed that placement of higher-order layers significantly impacts performance (Appendix A.3), with deeper placement leading to performance degradation. Finally, our current implementation does not employ low-rank approximations or other efficiency techniques commonly used in modern architectures, representing an opportunity for future optimization.

### 5.2 Future Directions

An important consideration is how our higher-order convolution approach relates to Vision Transformers [42], which have emerged as powerful alternatives to CNNs. While both capture feature interactions, they operate on fundamentally different principles. Our higher-order convolutions explicitly model multiplicative interactions between neighboring pixels through polynomial expansions, whereas transformers employ bilinear operations with global attention. From a mathematical perspective, attention mechanisms compute pairwise interactions through the query-key product ($QK^T$), which are inherently bilinear, while our approach extends to trilinear and higher-order multiplicative terms within local receptive fields. These differences suggest that higher-order convolutions and vision transformers may be complementary rather than competing approaches, potentially offering different advantages for capturing local complex patterns versus global relationships.

A particularly promising direction involves hybrid architectures that leverage both mechanisms: higher-order convolutions could process local complex patterns and higher-order correlations in early layers, while attention mechanisms capture long-range dependencies and global context in deeper layers. This combination could exploit the strengths of both approaches—efficient local feature extraction through higher-order polynomial expansions and flexible global reasoning through self-attention. Investigating such architectures could reveal whether these mechanisms provide complementary or redundant information, and identify optimal integration strategies for different vision tasks.

Beyond architectural hybridization, future research directions include exploring deeper architectures with multiple higher-order convolutional layers and investigating efficiency improvements through low-rank tensor decomposition methods. These techniques, which have proven successful in transformer architectures [43], could maintain the benefits of higher-order interactions while significantly reducing computational overhead. Additionally, extending our approach to other computer vision tasks beyond image classification, such as object detection and dense prediction tasks, represents an important avenue for validating the generalizability of higher-order convolutions.

In conclusion, our higher-order CNN represents a promising direction for more effective and biologically inspired computer vision models. By exploiting visual patterns and relationships more efficiently, even in shallow architectures, our approach opens new possibilities for advancing the field. Future research should focus on understanding the interplay between different orders of nonlinearity, their relationship with attention mechanisms, and investigating applications across diverse vision tasks.

## Acknowledgments

We would like to thank the reviewers for their valuable feedback and suggestions that helped improve this work. We thank Jan Koenderink and Andrea van Doorn for valuable discussions and for their detailed explanations of their work, which has been a source of inspiration for this research. We are grateful to Felix Wichmann for early discussions during the preliminary stages of this work. We also thank the Sorbonne Center for Artificial Intelligence (SCAI) for providing essential computational resources.

Simone Azeglio acknowledges this work was done within the framework of the PostGenAI@Paris project with the reference ANR-23-IACL-0007. Additionally, his PhD position was funded by the SCAI, through IDEX Sorbonne Université, project reference ANR-11-IDEX-0004.

Olivier Marre would like to thank the ERC Consolidator grant DEEPRETINA (101045253) and the Agence Nationale de la Recherche (ANR) for financial support through Chaire Industrielle MyopiaMaster (ANR-22-CHIN-0006), ANR-20-CE37-0018-04-Shooting Star, ANR-22-CE37-0033 NUTRIACT, ANR-22-CE37-0016-01 PerBaCo, and ANR-RetNet4EC.

Peter Neri acknowledges support from IIT (grant IVXX000701), the Agence nationale de la recherche (ANR-10-LABX-0087, ANR-10-IDEX-0001-02, ANR-22-CE93-0013), and CNRS.

Ulisse Ferrari acknowledges this work was done within the framework of the PostGenAI@Paris project with the reference ANR-23-IACL-0007, and benefited from financial support by the ANR through grants NatNetNoise (ANR-21-CE37-0024) and IHU FOReSIGHT (ANR-18-IAHU-01). Our lab is part of the DIM C-BRAINS, funded by the Conseil Régional d'Île-de-France.

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

# A Technical Appendices and Supplementary Material

## A.1 Code Availability & Reproducibility

The code is available at https://github.com/sazio/HigherOrderConv.

## A.2 Computational costs versus traditional methods

We analyze the computational requirements of HoCNN versus traditional CNNs. Our HoCNN architecture for MNIST, Fashion MNIST and CIFAR-10 consists of:

- First layer: $3 \times 3$ Volterra kernels (8 channels) with first, second, and third-order interactions
- Two standard convolutional blocks (16 and 32 channels)
- Two fully connected layers with dropout ($p = 0.25$)

Total parameters: 80,262

In order to provide a comparison with a more complex CNN model, we additionally tested a deeper version of the Baseline CNN model. In terms of architecture the Deep CNN network consist:

- Eight convolutional blocks ($3 \times 3$ kernels) with channel progression: $8 \to 8 \to 16 \to 16 \to 32 \to 32 \to 64 \to 64$
- Two fully connected layers with dropout ($p = 0.25$)

Total parameters: 108,130 (25% more than HoCNN)

Despite fewer parameters, HoCNN achieves superior accuracy (avg 72.35% vs 71.20%).

**Computational complexity scaling:**

- Standard convolution: $\mathcal{O}(C_{out} \times C_{in} \times K^2 \times H \times W)$
- Second-order kernels: $\mathcal{O}(C_{out} \times C_{in} \times K^4 \times H \times W)$
- Third-order kernels: $\mathcal{O}(C_{out} \times C_{in} \times K^6 \times H \times W)$

Where: $C_{out}$ = output channels, $C_{in}$ = input channels, $K$ = kernel size, $H, W$ = feature map height and width.

### A.2.1 FLOPs Measurements

To provide a concrete assessment of computational requirements, we conducted detailed benchmarking of our higher-order convolution operations compared to standard convolutions. For a single 3×3 convolutional layer processing an ImageNet-sized input:

- Order 1 (standard convolution): 0.2312 M FLOPs (baseline)
- Order 2: 1.1651 M FLOPs (5.04× standard convolution)
- Order 3: 4.3051 M FLOPs (18.62× standard convolution)

For complete networks on ImageNet-scale inputs:

- Standard ResNet-18: 1.82 G FLOPs
- HoResNet-18 (1st+2nd order): 2.64 G FLOPs (1.45× increase)
- HoResNet-18 (1st+2nd+3rd order): 3.00 G FLOPs (1.65× increase)

These measurements align with our theoretical complexity analysis while highlighting how strategic placement of higher-order operations primarily in early layers (see appendix:placementandlargerkernelscnn) keeps the overall computational increase manageable. Additionally, our implementation leverages symmetry in the higher-order kernels, significantly reducing the parameter count:

- Order 2: $1.80\times$ reduction ($(3 \times 3)^2 = 81 \to 45$ unique parameters)
- Order 3: $4.42\times$ reduction ($(3 \times 3)^3 = 729 \to 165$ unique parameters)

### A.2.2 FLOPs-Controlled Experiments

To isolate the impact of our higher-order convolution approach from simply having additional computational capacity, we conducted an experiment comparing our HoCNN with a deep CNN having comparable FLOPs on CIFAR-10:

Table 3: Comparison of accuracy with similar computational budgets

| Model | FLOPs | Accuracy (%) |
|-------|-------|--------------|
| Deep CNN | 7.09M | 71.20 |
| HoCNN | 6.81M | 72.06 |

These results confirm that the performance gains from higher-order convolutions cannot be attributed merely to increased computational capacity. Even with slightly fewer FLOPs, our HoCNN achieves a 1.21% accuracy improvement over the deep CNN. This substantiates our claim that higher-order convolutions provide a meaningful inductive bias that enables networks to learn more effective representations than those captured by simply scaling standard architectures.

### A.3 Placement of Higher-Order Layer & Effects of Larger Kernels in CNN

We conducted extensive experiments to investigate both the optimal placement of higher-order operations within the network hierarchy and the effects of increased kernel sizes in standard CNNs. These experiments provide important insights into the architectural choices made in our main study.

#### A.3.1 Effect of Higher-Order Layer Placement

We tested the introduction of the higher-order block at different depths in the network, specifically at the second and third convolutional layers. Our results reveal a consistent pattern across datasets: performance gradually degrades as the higher-order block is placed deeper in the architecture. For CIFAR-10, accuracy decreases from 71.17 ± 0.51% when the higher-order block is at the second convolutional layer to 70.29 ± 0.49% at the third layer. Similarly, for CIFAR-100, accuracy drops from 37.27 ± 0.80% to 36.09 ± 0.81%.

This performance degradation can be attributed to earlier layers potentially capturing and overfitting to spurious correlations. While standard convolutions theoretically preserve higher-order correlations, the cascade of nonlinearities and pooling operations can introduce and amplify spurious correlations, potentially compromising the network's ability to learn meaningful representations at deeper layers.

#### A.3.2 Impact of Kernel Size

We also investigated whether simply increasing kernel sizes in standard CNNs could achieve benefits similar to our higher-order operations. Despite larger kernels theoretically being capable of capturing higher-order correlations (as demonstrated by improved performance on our synthetic texture dataset), experiments with 5×5 and 7×7 kernels showed decreased test accuracy compared to our baseline CNN. For CIFAR-10, accuracy decreased to 68.72 ± 0.63% with 5×5 kernels and 67.51 ± 0.61% with 7×7 kernels. The pattern was similar for CIFAR-100, where accuracy dropped to 36.04 ± 0.76% with 5×5 kernels and 34.01 ± 0.72% with 7×7 kernels.

Our choice of 2×2 kernels for the main experiments was motivated by the generative process of our synthetic textures, which are based on gliders up to 4 pixels arranged in 2×2 patches. The results suggest that while larger kernels might have the capacity to capture higher-order interactions, the linear nature of standard convolutions makes it challenging to effectively learn these patterns during optimization.

#### A.3.3 Computational Considerations

The placement of higher-order operations in early layers offers computational advantages. In typical convolutional architectures, the number of channels increases with network depth. Introducing higher-order kernels in later layers, where channel count is larger, would substantially increase both parameter count and computational complexity. Our architectural choice of early placement thus provides an effective balance between computational efficiency and performance.

These findings support our design decision to introduce higher-order operations early in the network, where they can directly process raw image statistics before potential distortion by successive processing layers. The results demonstrate that explicitly modeling higher-order interactions through dedicated operations provides a more effective approach for capturing and leveraging complex statistical features compared to simply increasing kernel sizes in standard CNNs.

## A.4 ResNet Scaling Experiments

To thoroughly investigate whether the benefits of our higher-order approach extend to deeper architectures, we conducted comprehensive scaling experiments with ResNet-18, ResNet-34, and ResNet-50 architectures. While our main experiments focused on ResNet-18, these additional experiments provide valuable insights into how higher-order convolutions perform as model capacity increases.

The results comparing standard ResNet and HoResNet across different depths are presented in the following table:

Table 4: Performance comparison of ResNet vs. HoResNet across architectures of increasing depth

| Dataset | Standard ResNet | | | Higher-Order ResNet | | |
|---|---|---|---|---|---|---|
| | *R18* | *R34* | *R50* | *HoR18* | *HoR34* | *HoR50* |
| MNIST | $99.34 \pm 0.10$ | $99.35 \pm 0.11$ | $99.38 \pm 0.10$ | $99.38 \pm 0.09$ | $99.40 \pm 0.08$ | $\mathbf{99.41 \pm 0.08}$ |
| FashionMNIST | $90.85 \pm 0.32$ | $91.15 \pm 0.23$ | $91.20 \pm 0.24$ | $91.25 \pm 0.29$ | $91.32 \pm 0.21$ | $\mathbf{91.34 \pm 0.25}$ |
| CIFAR-10 | $81.26 \pm 0.57$ | $83.02 \pm 0.64$ | $84.10 \pm 0.59$ | $83.05 \pm 0.55$ | $84.06 \pm 0.49$ | $\mathbf{85.50 \pm 0.62}$ |
| CIFAR-100 | $60.77 \pm 0.73$ | $58.05 \pm 0.69$ | $56.46 \pm 0.83$ | $\mathbf{61.85 \pm 0.81}$ | $58.35 \pm 0.70$ | $56.55 \pm 0.74$ |

The results clearly demonstrate that the benefits of our higher-order approach extend beyond ResNet18. Key observations include:

- **Consistent improvements across model capacities:** Our higher-order networks consistently outperform their standard counterparts across all tested architectures and datasets. This suggests that the higher-order interactions captured by our approach provide fundamental benefits regardless of network depth.

- **Scaling behavior:** While both standard and higher-order networks show diminishing returns with increasing depth on simpler datasets (as expected), our HoResNets maintain their advantage. This indicates that the inductive bias provided by higher-order convolutions remains valuable even as models scale up.

- **Performance on complex datasets:** For CIFAR-10, our HoResNet18 achieves accuracy comparable to ResNet34, while HoResNet50 shows the strongest performance overall. This demonstrates the complementary benefits of higher-order operations and increased network depth.

- **Overfitting considerations:** We observe some evidence of overfitting as models get larger, particularly for CIFAR-100, where the limited number of samples per class makes training very deep networks challenging. This explains why we initially focused on more compact architectures in our main experiments.

These findings strongly support our claim that higher-order convolutions provide meaningful improvements that complement traditional scaling strategies like increasing network depth.

## A.5 Corruption Robustness Analysis

To evaluate the practical robustness of our higher-order models to real-world image corruptions, we conducted comprehensive experiments using the CIFAR-10-C and CIFAR-100-C benchmarks. These datasets apply 17 common image corruptions at 5 levels of severity to the standard test sets. We report results in terms of normalized accuracy (with respect to performance on the uncorrupted test set).

Our results reveal a surprising pattern: despite exhibiting increased sensitivity to our controlled higher-order statistical perturbations (Section 4), HoCNN models demonstrate enhanced robustness to common image corruptions:

Table 5: Normalized accuracy (%) on CIFAR-10-C with severity level 1

| Corruption Type | Baseline CNN | Higher-Order CNN |
|---|---|---|
| brightness | 99.4 ± 9.2 | **99.6 ± 5.2** |
| contrast | 96.0 ± 14.7 | **96.8 ± 6.5** |
| defocus_blur | 98.8 ± 9.2 | **99.1 ± 7.0** |
| elastic_transform | 85.3 ± 13.3 | **86.1 ± 8.5** |
| fog | 97.8 ± 12.9 | **98.6 ± 6.8** |
| frost | 91.2 ± 24.8 | **92.9 ± 13.5** |
| gaussian_noise | 76.2 ± 77.2 | **77.5 ± 30.0** |
| glass_blur | **58.2 ± 86.7** | 56.6 ± 59.0 |
| impulse_noise | 88.3 ± 25.3 | **89.6 ± 11.6** |
| jpeg_compression | 96.3 ± 12.6 | **97.0 ± 6.0** |
| motion_blur | **88.6 ± 11.2** | 88.4 ± 15.7 |
| pixelate | **97.6 ± 11.1** | 97.2 ± 5.9 |
| saturate | 99.5 ± 10.4 | **99.9 ± 5.0** |
| shot_noise | 85.8 ± 50.4 | **87.5 ± 18.9** |
| snow | **90.8 ± 28.1** | 90.8 ± 8.8 |
| spatter | 95.0 ± 12.8 | **95.5 ± 3.1** |
| zoom_blur | 81.5 ± 30.1 | **82.5 ± 18.0** |

Table 6: Normalized accuracy (%) on CIFAR-100-C with severity level 1

| Corruption Type | Baseline CNN | Higher-Order CNN |
|---|---|---|
| brightness | 99.1 ± 9.8 | **99.2 ± 12.3** |
| contrast | 93.9 ± 12.6 | **94.9 ± 11.5** |
| defocus_blur | 97.9 ± 11.3 | **98.2 ± 12.4** |
| elastic_transform | 75.1 ± 13.3 | **75.6 ± 10.0** |
| fog | 96.3 ± 10.3 | **97.0 ± 11.0** |
| frost | 84.8 ± 10.3 | **86.0 ± 11.8** |
| gaussian_noise | 60.5 ± 18.6 | **64.7 ± 16.6** |
| glass_blur | **34.6 ± 21.9** | 33.2 ± 20.0 |
| impulse_noise | 72.0 ± 19.8 | **73.1 ± 15.6** |
| jpeg_compression | 92.1 ± 10.9 | **93.5 ± 10.8** |
| motion_blur | 80.7 ± 18.0 | **81.5 ± 12.3** |
| pixelate | 93.4 ± 14.3 | **94.0 ± 12.1** |
| saturate | **98.7 ± 11.1** | 98.6 ± 11.6 |
| shot_noise | 74.0 ± 16.8 | **78.3 ± 15.4** |
| snow | 81.0 ± 13.3 | **82.3 ± 12.8** |
| spatter | 88.5 ± 13.4 | **89.4 ± 11.4** |
| zoom_blur | 73.0 ± 21.5 | **74.2 ± 14.5** |

- For CIFAR-10-C, our HoCNN outperforms the baseline on 13 out of 17 corruptions at severity level 1, with similar patterns observed at higher severity levels.

- For CIFAR-100-C, HoCNN shows even more pronounced improvements, outperforming the baseline on 15 out of 17 corruptions at severity level 1.

- The improvements are particularly notable for corruptions that preserve important image structure (contrast, blur, fog) while maintaining competitive performance on noise-based corruptions.

These findings demonstrate that while the higher-order networks are more sensitive to specific statistical manipulations that target their core functionality, they generally exhibit improved robustness to common image corruptions encountered in real-world scenarios. This suggests that the richer representational capabilities of higher-order convolutions may actually enhance model reliability in practical applications.

### A.6 Imagenette Results

To investigate the scalability of our approach on a moderately-sized dataset, we evaluated HoResNet-18 on Imagenette [44], a 10-class subset sampled from ImageNet. Imagenette provides a useful intermediate benchmark between the smaller CIFAR datasets and the full ImageNet, allowing for faster experimentation while maintaining the visual complexity of ImageNet images.

HoResNet-18 achieved 89.30% test accuracy compared to ResNet-18's 88.13%, demonstrating a +1.17% improvement (**Table** 7). Additionally, our HoResNet-18 architecture achieves superior performance compared to a ResNet-18 adapted VOneNet [45], a biologically-inspired model, while utilizing fewer parameters, as detailed in **Appendix** A.10.

Table 7: Imagenette classification accuracy

| Model | Accuracy (%) | # Params |
|---|---|---|
| ResNet-18 | 88.13 | 11,181,642 |
| HoResNet-18 | 89.30 | 11,168,382 |

*Training Details.* For Imagenette, images were normalized using mean = [0.5, 0.5, 0.5] and standard deviation = [0.5, 0.5, 0.5], in order to align with VOneNet [45] preprocessing. All other training details remain consistent with the benchmarks presented in Section 3.2. The Imagenette experiments were conducted on an NVIDIA RTX 4080 Ti GPU with training times of approximately 4-5 hours per model.

Due to computational constraints, we report single-run results for Imagenette rather than the multiple-run analysis performed for other benchmarks. While this limits our ability to assess variance, the improvement is consistent with the trends observed across other datasets, particularly the +0.85% improvement on the full ImageNet with statistical significance confirmed across 5 runs (Section 3.2).

### A.7 Fine-Grained Classification: CUB-200-2011

To further evaluate our approach on tasks requiring fine-grained visual discrimination, we tested HoResNet-18 on the CUB-200-2011 bird species classification dataset [**?** ]. Fine-grained classification tasks are particularly relevant for assessing higher-order feature extraction, as they require models to capture subtle visual patterns that distinguish between visually similar categories. The CUB-200-2011 dataset contains 11,788 images across 200 bird species, with images exhibiting significant intra-class variation in pose, scale, and background.

HoResNet-18 achieved 72.31% test accuracy compared to ResNet-18's 68.24%, demonstrating a substantial improvement of +4.07% (**Table** 8). This larger improvement compared to standard ImageNet classification (+0.85%) suggests that higher-order correlations play an increasingly important role in tasks requiring discrimination of subtle visual patterns. The results align with our hypothesis that natural images contain higher-order statistical structures that are particularly relevant for fine-grained recognition tasks.

Table 8: Fine-grained classification accuracy on CUB-200-2011

| Model | Accuracy (%) | # Params |
|---|---|---|
| ResNet-18 | 68.24 | 11,689,512 |
| HoResNet-18 | 72.31 | 11,676,252 |

*Training Details.* For CUB-200-2011, we followed standard fine-grained classification protocols. Images were resized to 224×224 pixels and normalized using ImageNet statistics (mean = [0.485, 0.456, 0.406], std = [0.229, 0.224, 0.225]). The training procedure matched our ImageNet setup: Stochastic Gradient Descent with momentum = 0.9, initial learning rate = 0.1 with cosine annealing, weight decay = 1e-4, and batch size = 128. We used standard data augmentation including random crops, horizontal flips, and color jittering. Models were trained for 100 epochs with the best model selected based on validation accuracy. Experiments were run on an NVIDIA A100 GPU, requiring approximately 12-18 hours per run.

Due to computational constraints, we report single-run results for CUB-200-2011 rather than the multiple-run analysis performed for ImageNet. However, the substantially larger improvement on this fine-grained task (+4.07% vs. +0.85% on ImageNet) provides strong evidence that higher-order convolutions are particularly beneficial for capturing subtle visual patterns.

## A.8 Fine-Grained Classification: Extended Analysis

The CUB-200-2011 dataset presents unique challenges for image classification models due to its fine-grained nature. Bird species often differ in subtle plumage patterns, beak shapes, body proportions, and coloration—visual features that require capturing complex spatial relationships beyond simple edges and basic textures. This makes CUB-200-2011 an ideal testbed for evaluating whether higher-order convolutions can capture the nuanced statistical structures necessary for fine-grained discrimination.

Our results on CUB-200-2011 show a +4.07% improvement (HoResNet-18: 72.31% vs. ResNet-18: 68.24%), which is notably larger than the +0.85% improvement observed on ImageNet. This differential improvement is particularly meaningful: while ImageNet contains diverse object categories with substantial inter-class visual differences, CUB-200-2011 requires discriminating between visually similar bird species. The larger performance gain on CUB-200-2011 suggests that higher-order correlations become increasingly important as the visual discrimination task becomes more fine-grained.

The biological inspiration for our approach aligns well with this finding. Biological visual systems are known to excel at fine-grained discrimination tasks—for instance, humans can readily distinguish between similar bird species, fabric textures, or facial identities. Our incorporation of higher-order correlation detection mechanisms may capture some of these capabilities, explaining the particularly strong performance on fine-grained tasks.

**Implementation Details:** We followed standard fine-grained classification protocols for CUB-200-2011. Images were resized to 224×224 pixels and normalized using ImageNet statistics (mean = [0.485, 0.456, 0.406], std = [0.229, 0.224, 0.225]). The training procedure matched our ImageNet setup: Stochastic Gradient Descent with momentum = 0.9, initial learning rate = 0.1 with cosine annealing, weight decay = 1e-4, and batch size = 128. We used standard data augmentation including random crops, horizontal flips, and color jittering. Models were trained for 100 epochs with the best model selected based on validation accuracy.

**Error Analysis:** Examining the confusion patterns, we observed that HoResNet-18 particularly excelled at distinguishing species with subtle plumage pattern differences. The most common errors for both architectures occurred between species with similar body shapes and coloration but different fine-scale patterns—precisely the scenarios where higher-order correlations would be expected to provide the most benefit. This error pattern supports our hypothesis that higher-order convolutions enhance the model's ability to capture the complex spatial relationships that define fine-grained visual categories.

## A.9 Architectures and Training Procedure: MNIST, FashionMNIST, CIFAR-10, CIFAR-100 & Imagenette

For image classification benchmarks, we employed a baseline CNN and a HoCNN architecture, with varying complexity depending on the dataset. For MNIST, FashionMNIST, and CIFAR-10, we used a simple architecture consisting of three convolutional blocks (Convolutional Layer, Batch Normalization, ReLU nonlinearity, and Max pooling) followed by two fully connected layers interleaved with dropout and ReLU nonlinearity (see **Figure** 3). For these datasets, we used 9, 18, and 36 kernels respectively for the convolutional layers of the baseline CNN, while the HoCNN used 8, 16, and 32 kernels. This configuration ensured that the baseline CNN had a comparable, slightly higher number of parameters (82,706 vs. 80,262). For CIFAR-100, we increased the kernel counts to 34, 68, and 136 for the baseline CNN, and 32, 64, and 128 for the HoCNN, resulting in 385,006 and 383,478 parameters respectively.

The validation accuracy curves for the CNN and HoCNN on these benchmarks are provided in the **Figure** 5, further illustrating the consistent performance advantage of the HoCNN across datasets.

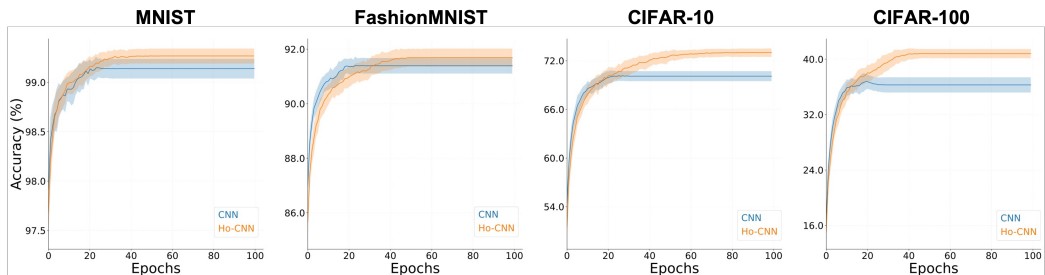

Figure 5: **Validation accuracy curves for CNN and HoCNN on image classification benchmarks**. Learning curves comparing the performance of the baseline CNN and the proposed HoCNN on MNIST, FashionMNIST, CIFAR-10, and CIFAR-100 datasets. The HoCNN consistently outperforms the CNN across all benchmarks, demonstrating the effectiveness of incorporating higher-order interactions in the convolutional layers.

Our training strategy utilized the AdamW optimizer with weight decay and learning rate scheduling (ReduceLROnPlateau). We employed Early Stopping with patience for model selection. For MNIST, FashionMNIST, CIFAR-10, and CIFAR-100, we trained 50 model realizations with different random seeds to ensure reliable performance assessment. For Imagenette, we conducted a single comprehensive evaluation run.

### A.9.1 Architectures and Parameter Counts: Imagenette

Standard ResNet-18 architecture, totaling 11,181,642 parameters:

Initial Block:

- Convolutional layer ($7 \times 7$ kernels, 64 channels)
- Batch normalization
- ReLU activation
- Max pooling ($3 \times 3$)

Four main stages, each containing 2 residual blocks:

- Stage 1: 64 channels (2 blocks)
- Stage 2: 128 channels (2 blocks)
- Stage 3: 256 channels (2 blocks)
- Stage 4: 512 channels (2 blocks)

Final layers:

- Global average pooling
- Fully connected layer to output classes

Higher-Order ResNet-18 (HoResNet-18), totaling 11,168,382 parameters:

Initial Block (similar to ResNet-18):

- Convolutional layer ($7 \times 7$ kernels, 30 channels)
- Batch normalization
- ReLU activation
- Max pooling ($3 \times 3$)

Four main stages, with a hybrid approach:

- Stage 1: 30 channels with higher-order residual blocks (2 blocks)

- Stage 2: 128 channels with standard residual blocks (2 blocks)

- Stage 3: 256 channels with standard residual blocks (2 blocks)

- Stage 4: 512 channels with standard residual blocks (2 blocks)

Final layers (similar to ResNet-18):

- Global average pooling

- Fully connected layer to output classes

### A.9.2   Training Setup: Imagenette

Data Preprocessing:

- Training images: Random resized crop to $224 \times 224$, random horizontal flip, normalization

- Test images: Resize to $256 \times 256$, center crop to $224 \times 224$, normalization

- Both models use normalization with mean and std of [0.5, 0.5, 0.5] which results in rescaled pixels with mean close to zero ([-0.07, -0.09, -0.15]) and std close to 0.5 ([0.55, 0.54, 0.58])

Training Configuration:

- Batch size: 64

- Loss function: Cross-entropy

- Optimizer: AdamW with learning rate 0.001 and weight decay 5e-4

- Learning rate scheduling: ReduceLROnPlateau (halves LR after 5 epochs without improvement)

- Early stopping: Implemented with 12 epochs patience

### A.10   Comparison with VOneNet

We compared the performance of our HoResNet-18 architecture with a ResNet-18 adapted VOneNet [45], a biologically-inspired model designed to better capture the properties of the visual cortex. Our HoResNet-18 model achieved superior accuracy on the Imagenette dataset (89.30%) compared to the VOneNet model (88.02%), while utilizing fewer parameters (11,168,382 vs. 14,445,322).

This improved performance aligns with the findings reported by the VOneNet authors, who observed lower ImageNet accuracy compared to standard ResNet architectures, despite achieving higher brain-scores (i.e., higher explained variance of neural activity recorded from specific brain areas in the visual cortex). This trade-off between ImageNet performance and brain-score can be verified on the Brain-Score framework website by sorting models based on their ImageNet top-1 accuracy.

Our results demonstrate that the HoResNet-18 architecture, equipped with higher-order convolutions, can outperform biologically-inspired models like VOneNet on standard image classification tasks while maintaining a more compact parameter footprint.

### A.11   Tied-Weights Issue

A depiction of how Higher-order terms go beyond simple non-pointwise nonlinearities is provided in **Figure** 6 while PCA analysis with different nonlinearities is provided in **Figure** 7, obtaining similar qualitative results as with ReLU.

### A.12   Sensitivity to Image Statistics

More detailed results are presented in **Table** 9 and 10 (in **bold** largest decrease in performance across models).

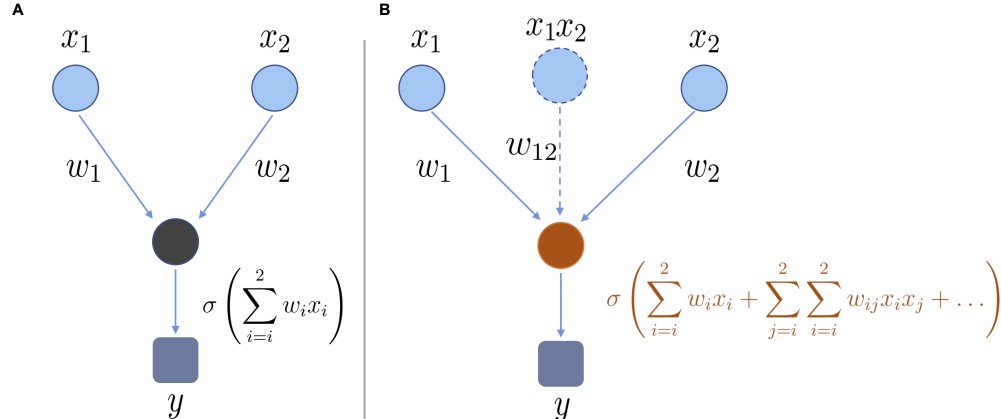

Figure 6: **Beyond pointwise nonlinearities** (A) Classical pointwise nonlinearity: the nonlinear function is applied after summing the input from the previous layer. A Taylor expansion of this non-linearity reveals the *tied-weight* problem as addressed in **Subsection** 1.2. (B) Non-pointwise non-linearity includes the summation of quadratic or higher-order terms. By modulating the interactions between terms we can untie weights in the Taylor expansion: the optimization process is not saddled with discovering an efficient trade-off between good approximation and balance of different terms.

Table 9: CNN Model accuracy (%) for each perturbation type and for different intensities I, in parentheses the decrease in accuracy with respect to the case without perturbation (CIFAR-10 avg test accuracy = 69.76%).

| Perturbation | I=0.05 | I=0.09 | I=0.12 | I=0.16 | I=0.20 |
|---|---|---|---|---|---|
| $\gamma$ | 67.0 (**-4.0%**) | 59.9 (-14.1%) | 50.5 (-27.6%) | 41.4 (-40.7%) | 34.1 (-51.2%) |
| $\beta$- | 66.5 (-4.7%) | 59.7 (-14.5%) | 50.4 (-27.7%) | 40.2 (-42.3%) | 31.2 (-55.2%) |
| $\beta$\| | 57.8 (-17.1%) | 41.5 (-40.5%) | 30.8 (-55.8%) | 24.6 (-64.7%) | 21.0 (-69.9%) |
| $\beta$\ | 55.5 (**-20.5%**) | 31.2 (-55.2%) | 18.0 (-74.2%) | 13.2 (-81.0%) | 11.8 (-83.1%) |
| $\beta$/ | 56.6 (-18.9%) | 32.6 (-53.2%) | 19.0 (-72.8%) | 13.9 (-80.1%) | 12.2 (-82.5%) |
| $\theta_{\llcorner}$ | 54.2 (-22.2%) | 30.9 (-55.6%) | 18.6 (-73.3%) | 13.8 (-80.2%) | 11.9 (-82.9%) |
| $\theta_{\ulcorner}$ | 54.4 (-22.0%) | 31.2 (-55.3%) | 18.7 (-73.1%) | 13.9 (-80.1%) | 11.9 (-82.9%) |
| $\theta_{\urcorner}$ | 54.3 (-22.1%) | 31.0 (-55.6%) | 18.8 (-73.1%) | 13.9 (-80.0%) | 12.0 (-82.7%) |
| $\theta_{\lrcorner}$ | 54.2 (-22.3%) | 30.7 (-55.9%) | 18.5 (-73.5%) | 13.7 (-80.3%) | 11.9 (-83.0%) |
| $\alpha$ | 56.1 (-19.6%) | 36.0 (-48.4%) | 24.0 (-65.7%) | 18.3 (-73.7%) | 15.8 (-77.3%) |

## A.13 Neural Representations: Representational Similarity Analysis

To gain deeper insight into how our Higher-order Convolutional layer (HoConv) processes information differently from a standard Convolutional (Conv) layer, we employed Representational Similarity Analysis (RSA) [17] on the CIFAR-10 dataset. We analyzed activations from 100 test images (10 per class) averaged across 50 model realizations to mitigate the effects of random initialization [46]. Our findings reveal distinct representational geometries between the two models. In **Figure** 8 we consider an alternative distance (hellinger) between the two RDMs, based on the consideration of the fact that values are between 0 and 1.

### A.13.1 Order-wise Representational Differences

The 2$^{\text{nd}}$ and 3$^{\text{rd}}$ order components of the HoConv layer can be analyzed independently and extract different representations, with progressively smaller mean dissimilarity compared to the baseline Conv layer (**Figure** 11). These order-wise differences contribute to the overall representational differences between CNN and HoCNN.

Our analysis of neural representations revealed distinct patterns across different order components (see **Figure** 9 and **Figure** 11). The standard convolutional layer exhibited a unimodal distribution of pairwise distances, indicating a consistent representation of stimuli. In contrast, the 2$^{\text{nd}}$ order

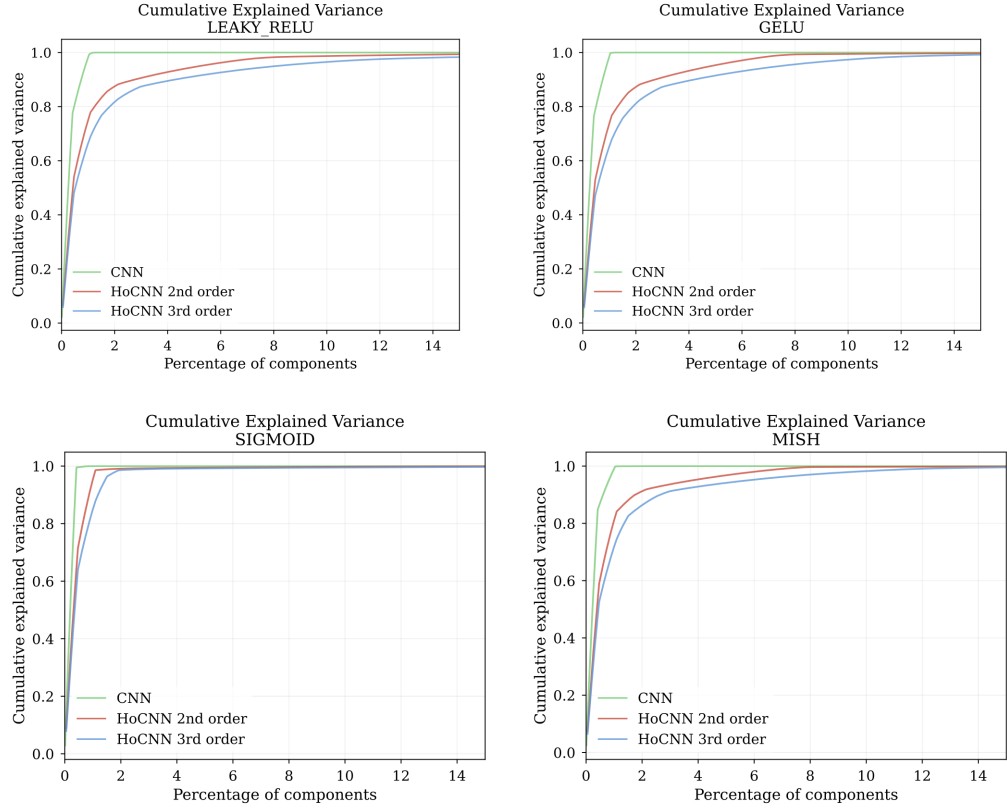

Figure 7: **PCA analysis with different nonlinearities: Leaky Relu, Gelu, Sigmoid, Mish**

Table 10: HO-CNN Model accuracy (%) for each perturbation type and for different intensities I, in parentheses the decrease in accuracy with respect to the case without perturbation (CIFAR-10 avg test accuracy = 72.87%).

| Perturbation | I=0.05 | I=0.09 | I=0.12 | I=0.16 | I=0.20 |
|---|---|---|---|---|---|
| $\gamma$ | 70.2 (-3.6%) | 62.5 (**-14.2%**) | 51.9 (**-28.8%**) | 41.2 (**-43.4%**) | 32.8 (**-55.0%**) |
| $\beta$- | 69.3 (**-4.9%**) | 61.7 (**-15.3%**) | 52.1 (**-28.5%**) | 41.3 (**-43.3%**) | 31.7 (**-56.5%**) |
| $\beta$\| | 58.5 (**-19.8%**) | 40.3 (**-44.8%**) | 29.1 (**-60.0%**) | 23.2 (**-68.2%**) | 20.2 (**-72.3%**) |
| $\beta$\ | 58.5 (-19.7%) | 30.6 (**-58.0%**) | 16.5 (**-77.4%**) | 12.6 (**-82.7%**) | 11.7 (**-84.0%**) |
| $\beta$/ | 58.8 (**-19.3%**) | 30.5 (**-58.2%**) | 16.4 (**-77.5%**) | 12.5 (**-82.8%**) | 11.7 (**-83.9%**) |
| $\theta_{\llcorner}$ | 56.3 (**-22.7%**) | 29.1 (**-60.1%**) | 16.0 (**-78.0%**) | 12.1 (**-83.4%**) | 10.9 (**-85.0%**) |
| $\theta_{\ulcorner}$ | 56.3 (**-22.8%**) | 29.1 (**-60.0%**) | 16.2 (**-77.8%**) | 12.2 (**-83.2%**) | 11.0 (**-84.9%**) |
| $\theta_{\urcorner}$ | 56.4 (**-22.6%**) | 29.4 (**-59.6%**) | 16.4 (**-77.5%**) | 12.3 (**-83.1%**) | 11.0 (**-84.9%**) |
| $\theta_{\lrcorner}$ | 56.3 (**-22.8%**) | 29.1 (**-60.1%**) | 16.1 (**-77.9%**) | 12.2 (**-83.3%**) | 11.0 (**-85.0%**) |
| $\alpha$ | 58.4 (**-19.8%**) | 35.8 (**-50.9%**) | 22.5 (**-69.2%**) | 17.0 (**-76.6%**) | 14.9 (**-79.6%**) |

component of the HoCNN showed a bimodal distribution, suggesting two distinct scales of representation. This bimodality could indicate the model's ability to capture both fine-grained similarities and broader categorical differences between stimuli.

The 3rd order component demonstrated a wider spread of distances, with a peak at lower dissimilarity values. This pattern suggests that the 3rd order component might be capturing more subtle relationships or features among the stimuli, complementing the representations of the lower-order components.

These findings point to a more nuanced and hierarchical representation in HoCNNs compared to standard CNNs. The multi-order structure appears to enable the model to simultaneously process information at various scales and levels of abstraction. This could potentially explain the enhanced

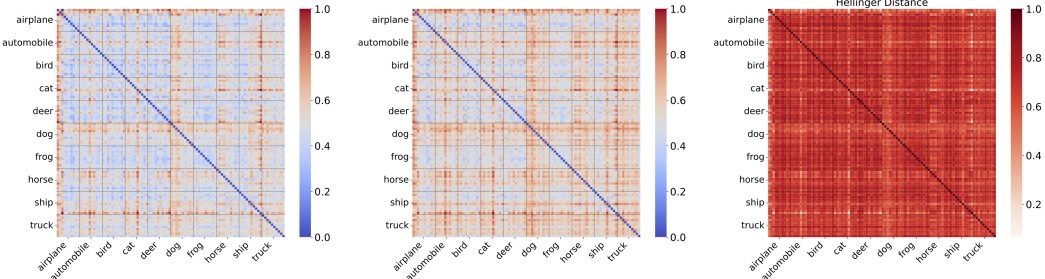

Figure 8: (Left) Representational Dissimilarity Matrix (RDM) for baseline Convolutional (Conv) Block (Conv layer, Batch Norm, ReLU, Pooling). (Middle) RDM for the Higher-order Convolutional (HoConv) block. (Right) Hellinger Distance between the two RDMs, capturing different representational geometries between the two blocks

performance of HoCNNs on certain tasks, as they may be better equipped to capture complex, multi-scale patterns in the input data.

However, further investigation is needed to fully understand the underlying mechanisms and implications of these representational differences. Future work could explore how these distinct representations contribute to specific aspects of model performance, and whether they align with known principles of biological visual processing.

### A.13.2    Layer-wise Representational Differences

The representational difference between CNN and HoCNN becomes more explicit when all orders are summed together and fed to the batch normalization, ReLU, and max pooling layers (see **Figure** 3), giving rise to the output of our HoConv block. We quantify these differences using Representational Dissimilarity Matrices (RDMs) on a subset of the testing set (**Figure** 4). We compare blocks instead of single layers because the standard convolution is linear, and equipping it with a nonlinearity layer makes the comparison fairer. The Conv and HoConv RDMs show clear structural differences (**Figure** 4A & B), with the HoConv exhibiting more pronounced class-specific patterns.

### A.13.3    Correlation between RDMs across Layers

We further analyzed the average correlation between RDMs across layers of the CNN and HoCNN (**Figure** 10). The results confirm the divergence in representational structure, with correlations decreasing in higher-order layers. This suggests that the HoConv layer progressively captures more diverse and complex features compared to the standard Conv layer.

### A.13.4    RSA on Texture Classification

We extended the analysis presented in paragraph 4 of the main paper to the four models trained on synthetic texture data. This analysis aims to highlight the contributions of different kernel orders in texture classification. Following a similar methodology, we randomly selected 10 samples for each texture class from the test set. We then extracted activations at specific layers of the models and used these to construct representational dissimilarity matrices (RDMs), see **Figure** 12 , **Figure** 13and **Figure** 14.

These RDMs provide a visual representation of how the models distinguish between different texture classes at various processing stages. By comparing the RDMs across different layers and between the baseline CNN and HoCNN models, we can gain insights into how higher-order kernels influence the representation and discrimination of texture patterns.

The following figures present these RDMs, allowing for a comparative analysis of how information is processed and transformed through the network layers in both the standard CNN and the HoCNN architectures when dealing with synthetic texture data. This analysis complements our findings from natural image datasets and offers a more comprehensive view of the role of higher-order computations in visual processing tasks.

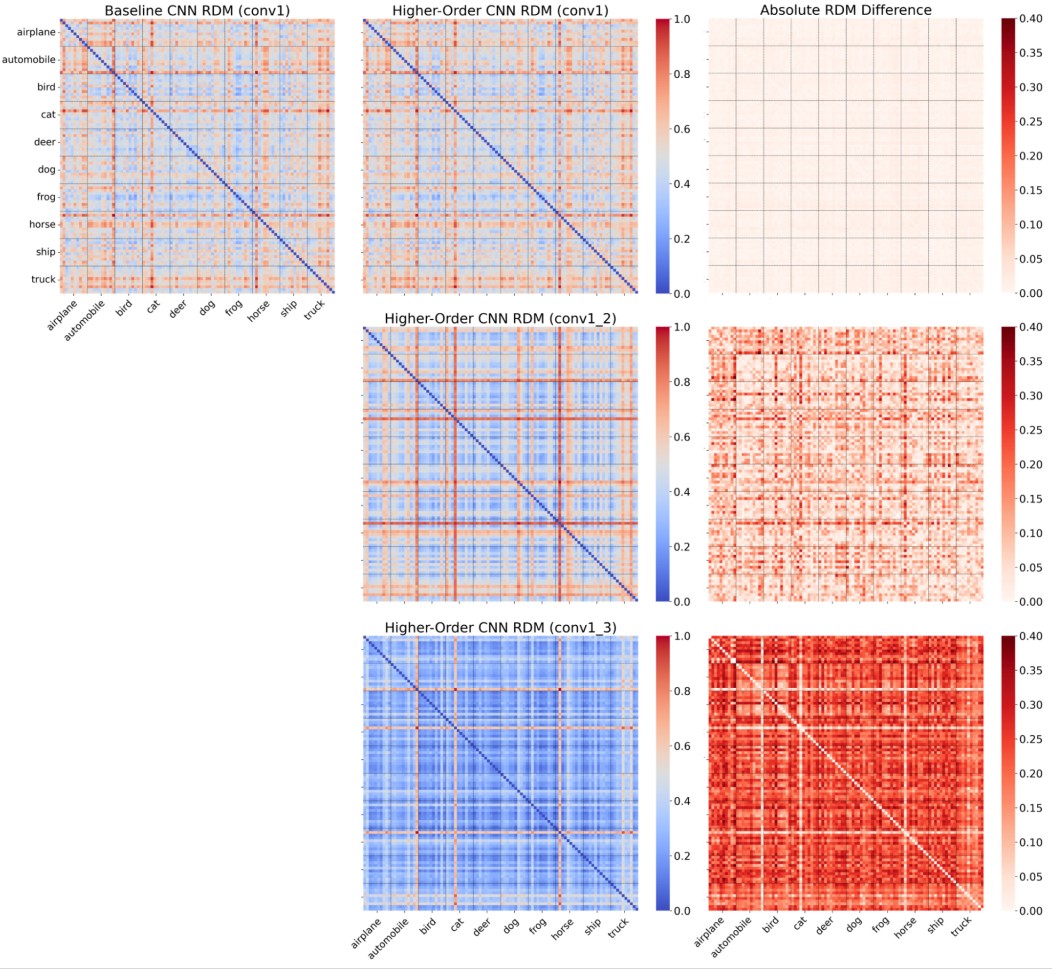

Figure 9: **Representational dissimilarity matrices across layers**. (First row) Representational Dissimilarity Matrices for (left) standard Conv layer (1st layer of the CNN), (middle) 1st order HoConv layer, (right) absolute difference between the two RDMs. (Second row), the RDM for standard Conv is not repeated while (in the middle) we show the RDM for the activations extracted after the 2nd order HoConv layer; (right) the absolute difference between the standard Conv and the 2nd order HoConv RDMs, which presents different representational geometries.

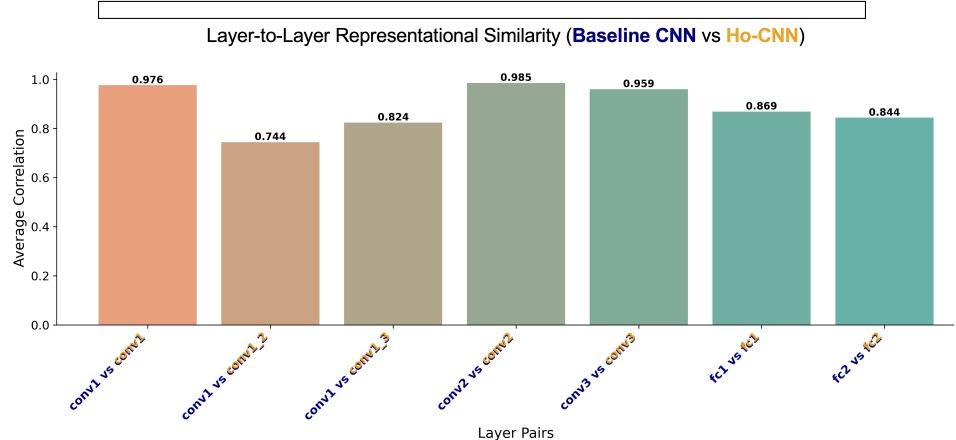

Figure 10: **Average Correlation between RDMs for matching layers (Baseline CNN vs HoCNN).**

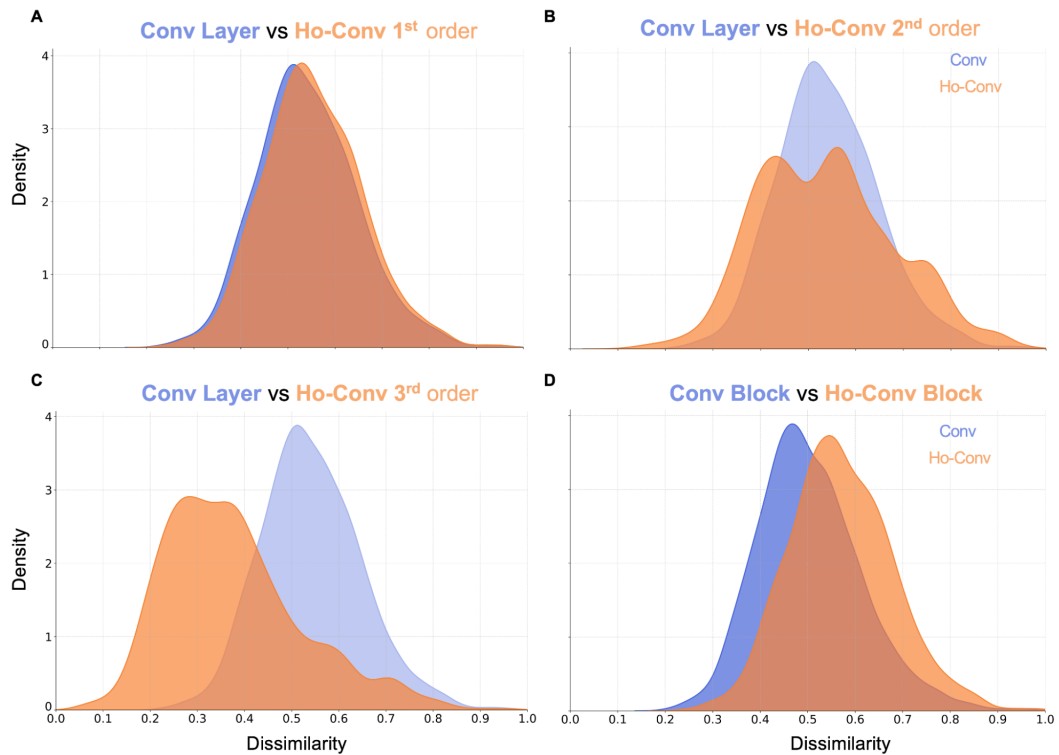

Figure 11: **Pairwise distance distribution of RDMs (CIFAR-10)**. An alternative way to represent RDM matrices is to plot the pairwise distance distribution of their entries. (A) We can see that at the 1st order the representations of the standard Conv Layer (here before the nonlinearity) and of the HoConv Layer are practically the same (1st order HoConv Layer is mathematically equivalent to a Conv Layer). (B) Representations for the 2nd order HoConv Layer are different from the standard convolutional layer: by themselves alone they don't bring better representations in terms of dissimilarity if compared with the standard Conv Layer ones. (C) Similar considerations for 3rd order HoConv Layer representations. (D) When considering the output of the two blocks indeed (same as Fig 5. D), the situation is different: the distribution corresponding to the HoConv representations is considerably shifted, leading to "more" dissimilar representations on average.

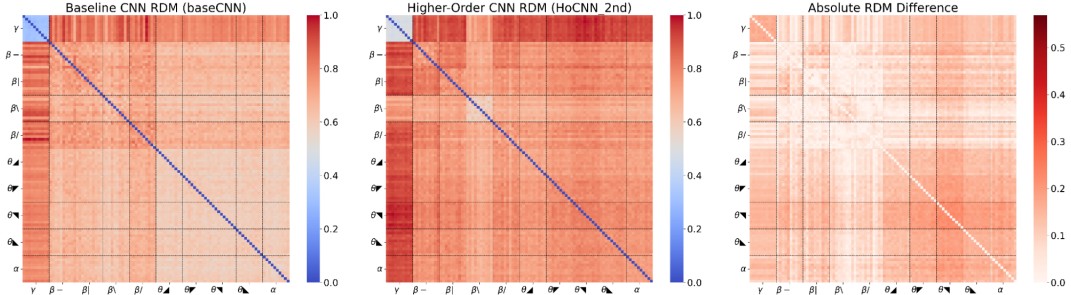

Figure 12: RDM comparison between baseline CNN and HoCNN 2nd order (summed 1st and 2nd order kernel), on the right absolute difference to highlight representational differences.

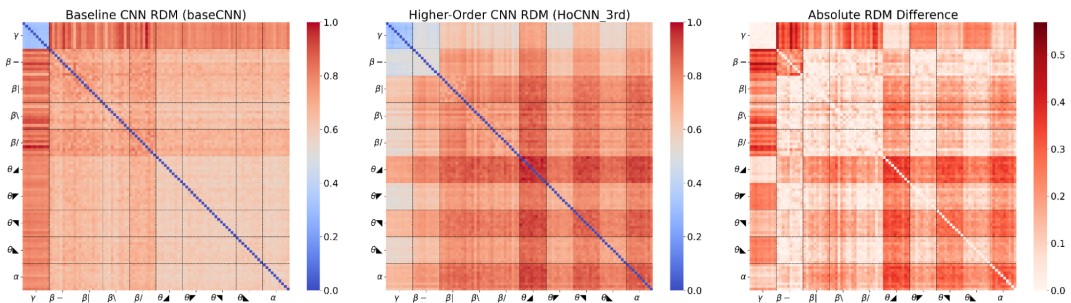

Figure 13: RDM comparison between baseline CNN and HoCNN 3$^{rd}$ order (summed 1$^{st}$, 2$^{nd}$ and 3$^{rd}$ order kernel), on the right absolute difference to highlight representational differences.

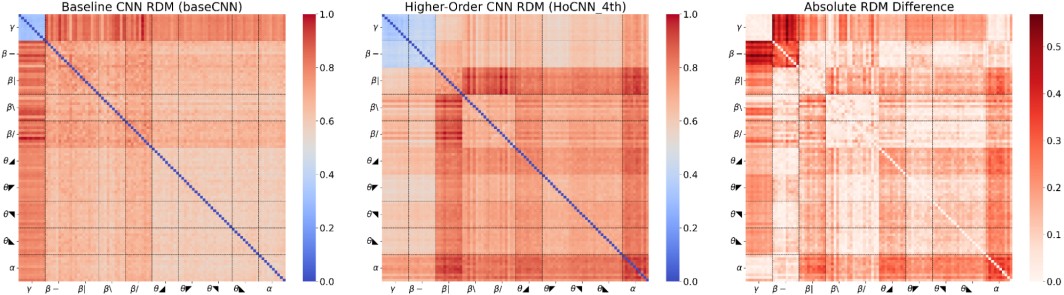

Figure 14: RDM comparison between baseline CNN and HoCNN 4$^{th}$ order (summed 1$^{st}$, 2$^{nd}$, 3$^{rd}$ and 4$^{th}$ order kernel), on the right absolute difference to highlight representational differences.

