# OpenReview forum: "Convolution Goes Higher-Order:  A Biologically Inspired Mechanism Empowers Image Classification"
_NeurIPS.cc/2025/Conference — NeurIPS 2025 poster_

### Official Review · Reviewer_Mz2V · 2025-06-07

**Clarity:** 2
**Significance:** 3
**Originality:** 4
**Rating:** 2
**Confidence:** 4

**Summary:**

The authors propose a novel convolution operator, which aims to solve the tied-weight issue. They intuitively derive this issue via a polynomial expansion of the classical convolution operator and address it by introducing higher orders of convolution into a single convolution layer. According to the paper, higher orders are enabled by the outer product of the two operands of the classical linear-weighted sum, along with an inner product. Experiments show the effectiveness of their method.

**Questions:**

**Line 90~91** -> ***solved***
> This tied-weight issue becomes particularly problematic when the network is not deep enough or wide enough to compensate...

On the one hand, does this mean that this method only shows significant benefit on shallow or narrow CNN models?
On the other hand, it seems necessary to provide experiment results to prove that shallow or narrow CNN models can be greatly augmented even beating relatively deep or wide CNN models.


**Figure 1** -> ***addressed***
> Figure 1.

Only spatial transformation is visualized, so how is the channel transformation implemented? I guess there must be channel transformation, i.e., not a depth-wise-like convolution, or the channel expansion can not be realized from 3, to 30, 128, 256 and 512.

Besides, what are the similarities between the authors' high-order design and the biological structure in the visual system? Providing this would help readers to follow your intuition a lot.


**Table 1** -> ***addressed***
> Table 1.

It would be better to clearly point out why the number of parameters of 2nd order is even lower than the baseline. As higher orders require much more learnable weights, readers would be doubtful about the numbers unless they dive deep into the appendix.


**Table 2** -> ***responded***
> Table 2.

The baseline performance, 68.67 of CNN (ResNet18) on ImageNet, is relatively too weak [1] and authors' method even is not as good as the recognized performance of the baseline, i.e., 69.9. It would be better that the authors run their experiments on some well-recognized codebase [1].


**Experiments** -> ***the work's effectiveness on segmentation is only validated on a very old toy dataset (agreed by the authors); those of detection is still not responded***
> Experiments

The experiments should cover those primary CV tasks, like object detection, instance segmentation or semantic segmentation, or the current reults can not fully support the authors' key claim.

> A.6 Comparison with VOneNet -> ***partly responded, not satisfying***

Comparision experiments with other bio-inspired methods should be placed in the main text. If more other bio-inspired methods, CNN methods like CoC [2] and even ViT methods like TransNeXt [3], are provided, it would make this work much more solid.


[1] https://github.com/open-mmlab/mmpretrain/tree/main/configs/resnet#image-classification-on-imagenet-1k

[2] Convolution of Convolution: Let Kernels Spatially Collaborate

[3] TransNeXt: Robust Foveal Visual Perception for Vision Transformers

**Ethical Concerns:**

["NO or VERY MINOR ethics concerns only"]

**Final Justification:**

Update 3
---
The authors still have not fully addressed my central concern. While I perceive their response as avoiding this core issue, a positive aspect is that their rebuttal has reinforced my assessment.



Update 2
---
I would like to re-acknowledge the Significance and Originality or this work. Yet the Quality still needs improvement after rebuttal.

As for primary cv tasks, the response is not satisfying. The authors use a dataset that is even older and easier than the toy dataset Pascal VOC 2012, let alone the classical benchmark MS COCO 2017. In CUB-200-2011, the dataset used by the authors, there is merely one object box/segment. in each image sample. This is too weak to truely test the effectiveness of a novel conv op in any modern CNN model. Just like in network pruning, some authors claim their algo can prune 99% of a CNN's parameters without undermining the accuracy on CIFAR10 -- Of course this is possible yet meaningless, because most of CV tasks require (pre-)training on ImageNet-scale datasets.

The results of detection is also not provided.

Please note that the three primary CV tasks are classification, detection and segmentation. All works that claim they propose an effective novel conv op should and they in fact have tested their method on those primary CV tasks -- Please kindly check the references mentioned above. Please also note that this is the key point supporting my final rating.

As for biological inspiration, the response is okay yet still provides too limited insightful analogy between the op and the bio to convince the readers that this is not a decoration of storytelling.

As for parameter efficiency, the response is good.

In short, I sincerely appreciate the authors' efforts in addressing my concerns and truly expect to see the authors providing the necessary results to make their novel work solid enough.



Update 1
---
Thank the authors for their efforts in rebuttal. I appreciate the revisions being made. However, after careful consideration, I still believe the initial rating accurately reflects the contribution of the work. I understand the effort involved in the revisions and the time constraints, and I hope my feedback will be helpful for future improvements.
Specifically, for example, experiments on those widely-recognized CV tasks should be MS COCO for detection and segmentation, instead of those outdated toy datasets.

**Limitations:**

Yes.

**Paper Formatting Concerns:**

NA.

**Quality:**

1

**Strengths And Weaknesses:**

Strengths
---

1. This work seems promising to me, but the experiments are too limited, just on a tentative level. It would lead to good significance to the community if the authors can supplement more concrete work, like well-design architectures with off-the-shelf pretrained model weights. Good examples are Res2Net, ConvMixer, ConvNeXt, DINO2, HorNet [1], and so on.

2. The authors derive the tied weight issue intuitively using polynomial expanding classifical convolution. To the best of my knowledge, there is no such work ever addressing such a problem.


Weaknesses
---

1. The quality of this paper is quite limited. Key claims are not fully addressed. For example, to fully prove the effectiveness of a novel operator, experiments on primary Computer Vision tasks should be covered, not just classification, also including detection and segmentation. Good examples are mentioned above.

=> ***The work's effectiveness of segmentation is only shown on a very old toy dataset (agreed by the authors); Those of detection has not been validated.***

2. The clarity is this paper is fair. Key designs of this method is clearly stated, but the corresponding biological inspiration is not, making the biological part a mere decoration of the storytelling. A good example could be CoC [2].

=> ***This point was repsonded but not satisfying, especially considering that the authors put the core claim of biological inspiration in their title.***

[1] https://github.com/open-mmlab/mmpretrain/tree/main/configs

[2] Convolution of Convolution: Let Kernels Spatially Collaborate.

---

> ### Author Rebuttal · Authors · 2025-07-31
>
> Dear Reviewer Mz2V,
>
> Thank you for recognizing significance and originality of our work. We appreciate your constructive feedback and address your concerns below.
>
>
> ### S1 off-the-shelf pretrained model
>
> We appreciate the reviewer’s suggestion to compare our approach with advanced architectures. In principle, integrating Higher-Order Convolutions into these state-of-the-art networks is an exciting direction. However, each of these models incorporates unique design optimizations – for example, Res2Net introduces multi-scale feature groups within residual blocks, ConvMixer relies on patch-level mixing of features, and ConvNeXt employs large convolutional kernels with specific normalization patterns. Adapting our higher-order convolution mechanism to each such architecture would require careful, case-by-case architectural tuning, effectively resulting in designing and training multiple new hybrid models. Albeit being very relevant future directions, our primary objective was to introduce the HoConv layer as a general concept and demonstrate its effectiveness in a controlled setting, using standard CNN backbones. We focused on baseline architectures to isolate the benefits of higher-order convolutions without the confounding factors introduced by complex architectural tricks. We believe this strategy provides clear evidence of our method’s merit. Expanding the evaluation to every cutting-edge architecture, while valuable, would significantly broaden the scope and detract from the core contribution of our work. We view such extensive comparisons and integrations as future work that extend beyond the space constraints and goals of the current paper.
>
> In summary, we agree that incorporating HoConv into modern architectures is an important next step, and we thank the reviewer for these suggestions. We will note this in the revised manuscript as a future direction. For now, we hope the results provided (see below for additional tasks) sufficiently validate the effectiveness of our proposed operator.
>
> ---
> ### W1 Experiments: Extension to primary CV tasks
>
> We agree with the reviewer that our HoConv layer can provide significant improvement also for other tasks in computer vision.  Following similar suggestions from Reviewer #3, we tested our approach on the CUB-200-2011 dataset (Wah et al 2011). This dataset, consisting of 200 bird species (classes) is one of the benchmarks of fine grained classification and also presents segmentation masks, it  is particularly challenging due to subtle inter-class differences and limited training data (~6k images), making it an excellent test case for our approach. We performed two tasks:
>
> * **Fine-grained classification**: Using, HoResNet18 showed larger improvements than on standard benchmarks (72.31 % +- 0.29% vs ResNet18’s 68.24 % +- 0.37%), validating that higher-order convolutions excel at capturing subtle discriminative patterns.
>
>
> * **Segmentation**: We developed a transposed higher-order convolution layer for decoder architectures, built a U-Net model using our pretrained encoder and used them in the segmentation task. The HoResNet18-based U-Net consistently outperformed the ResNet18 baseline in mean IoU (87.72 % +- 0.68% mIoU vs ResNet18’s 82.17 % +-  0.75%).
>
> These preliminary results are very encouraging and we will add them in the appendix, but we believe that an in depth analysis would be outside the scope of our paper. Here, we preferred to deepen the understanding of why/how our HoConv layer is effective in capturing higher order correlations in natural images, instead of testing multiple computer vision tasks beyond image classification.
>
> ---
> ### W2 Biological inspiration clarity
>
> We acknowledge that biological inspiration needs clearer articulation. A significant amount of literature (ex: Fitzgerald and Clark, eLife, 2015) has shown that neural circuits in early visual systems respond to higher order correlations, beyond pairwise. To implement this response, those circuits use non-linearities and products of more than two local filters applied to their inputs, as for example the multiplication between a low and a high-pass filter convoluted with one image patch with a high-pass convoluted with another patch. A standard convolution layer cannot reproduce those responses (Fitzgerald and Clark,  2015), and to mimic those behaviors CNNs need to stack multiple layers, going much deeper than the visual system itself. Inspired by these limitations of standard CNNs, we developed our higher-order convolution layer, and show that it can capture high order correlations present in the input image.
>
> After a first draft going deeper into the comparison between our model and early visual systems, we focused the paper on the ML architecture, removing those biological arguments from the manuscript to improve focus and readability. However the referee is right, as those insights can be helpful to understand the logic beyond our novelties,  we will add them back in a dedicated appendix in the camera-ready version. For this reason, we have extended relevant sections to provide more details regarding biological motivation, as suggested by the Reviewer. Additionally, forthcoming results in a companion paper will showcase how our HoConv layer helps improve prediction of neuronal responses across animal species to visual stimuli
>
> ---
> ### Q (Line 90-91): Shallow vs deep networks
>
> No, our HoConv layer is not beneficial only for shallow networks, as shown by our results on difficult image classification datasets (Table 2), where networks are deep.  Additionally, beyond the comparison with CNN matching the number of parameters, in App. A.1 we show that a shallow HoConv network can beat a deeper CNN with 25% more parameters.
> In summary, our claim isn't that the method only benefits shallow networks, but rather that it provides an efficient alternative to depth/width for capturing the complex correlation patterns present in natural images.
>
> ---
> ### Q (Figure 1): Channel transformation implementation
>
> * For each order, we have separate learnable weights of shape (out_channels, in_channels, kernel_space_size^order)
> * The input is unfolded into patches, and the higher-order kernel computes higher-order interactions within each patch
> * These are then multiplied by the weight matrix to perform channel transformation
> * All orders (1st, 2nd, 3rd, etc.) are summed to produce the final output
>
> Each order learns its own channel transformation weights (since different orders’ tensors have different shapes), allowing the network to discover which higher-order spatial patterns are most relevant for each output channel.
>
> For Biological inspiration see above, W2 response.
>
> ---
> ### Q (Table 1): Parameter count clarification
>
> We appreciate your concern about the parameter count, which is a crucial ingredient for appreciating HoConv networks. For the results reported in table 1, the CNN has more parameters than the 2nd order HoConv because its first layer is wider (10 channels for the CNN against 2 for the HoConv), while the subsequent layers have the same architecture.  This is explained in LL167-168 and LL172. We have chosen this setting exactly to show that a thin HoConv can outperform a wider CNN with more parameters. On the same line, as shown in App. A.1, HoConv can even outperform a deeper CNN with 25% more parameters.
>
> ---
> ### Q Table 2: ImageNet baseline performance
>
> We agree with the referee, our CNN baseline performance doesn’t match the SOTA by the open-mmlab codebase. The reason is that we trained both models from scratch - following the learning details provided in the manuscript - without the hyperparameters, preprocessing and augmentation tricks that have been designed to push performance at maximum. In our opinion this choice is necessary to allow a controlled comparison between CNN and HoConv networks: It would not have been fair to include those tricks developed for CNNs over the course of years, without doing the same for HoConv. We plan to work on incorporating those tricks in the future and generalize them also for HoConvs, but for the purpose of comparing CNN and HoConv on a level plain field it does not seem critical that they are included.
>
> **Experiments**  See response to W1 above
>
> ---
> ### Q Comparison with other bio-inspired architectures
>
> Thank you for suggesting our comparison with VOneNet be in the main text. While we agree this is an important comparison, as VOneNet is another bio-inspired convolutional model, we considered it a secondary result that complements our primary findings. Our main focus was to demonstrate the fundamental advantages of higher-order convolutions over standard CNNs, rather than comparing different bio-inspired methods. We will move this comparison as suggested, but retain its role as supporting evidence rather than a central claim.
>
> We appreciate you bringing “Convolution of Convolution” (CoC) to our attention—we were unaware of this work and will discuss it in the camera-ready version. Their architecture is not an alternative, but rather complementary to our HoConv layer, as HoConv layers could be integrated into CoC. For this reason, benchmarking HoConv against CoC is beyond our contribution’s scope.
>
> Regarding the TransNeXt comparison, we note that it is a Vision Transformer architecture, which is beyond our scope. Our primary goal was to provide a focused study on extending convolution operations, not to compare across fundamentally different paradigms. This focus allows us to isolate the specific benefits of higher-order operations with controlled comparisons.
>
> We believe our current experimental design, centered on controlled comparisons with standard CNNs, effectively demonstrates the benefits of our approach while keeping the scope manageable given our computational constraints.

---

> ### Author Response · Authors · 2025-08-05
>
> Dear Reviewer Mz2V,
>
> Thank you for acknowledging our rebuttal. We noticed that you've updated your confidence level to 3, which suggests you've carefully considered our response.
>
> Following the NeurIPS organizers' encouragement to continue discussions, we'd like to follow up on a few key points we addressed in our rebuttal:
>
> * **Extension to other CV tasks**: We provided results on both fine-grained classification and segmentation using CUB-200-2011, showing substantial improvements (HoResNet18 achieved +4.07% on classification and +5.55% on segmentation compared to ResNet18). Do these additional experiments address your concern about demonstrating effectiveness beyond image classification?
>
>
> * **Biological inspiration**: We've clarified how our approach relates to biological visual processing, particularly the higher-order correlation detection mechanisms found in early visual systems. We plan to expand this in an appendix for the camera-ready version. Does this adequately address your concern about biological motivation?
>
>
> * **Parameter efficiency**: We demonstrated that HoConv layers allow for more efficient network architectures, both in shallow networks (beating deeper CNNs with 25% more parameters) and in deeper architectures like ResNet18. Does our explanation of the parameter counts in Table 1 and the overall efficiency of our approach resolve your questions?
>
>
> We'd appreciate any further thoughts you might have on these points, especially regarding whether our experimental results sufficiently demonstrate the value of our approach.
>
> Thank you again for your time and consideration.

---

> > ### Comment · Reviewer_Mz2V · 2025-08-06
> >
> > I would like to re-acknowledge the Significance and Originality or this work. Yet the Quality still needs improvement after rebuttal.
> >
> > As for **primary cv tasks**, the response is not satisfying. The authors use a dataset that is even older and easier than the toy dataset Pascal VOC 2012, let alone the classical benchmark MS COCO 2017. In CUB-200-2011, the dataset used by the authors, there is merely one object box/segment. in each image sample. This is too weak to truely test the effectiveness of a novel conv op in any modern CNN model. **Just like in network pruning, some authors claim their algo can prune 99% of a CNN's parameters without undermining the accuracy on CIFAR10 -- Of course this is possible yet meaningless, because most of CV tasks require (pre-)training on ImageNet-scale datasets.**
> >
> > The results of detection is also not provided.
> >
> > Please note that the three primary CV tasks are classification, detection and segmentation. All works that claim they propose an effective novel conv op should and they in fact have tested their method on those primary CV tasks -- Please kindly check the references mentioned above. Please also note that this is the key point supporting my final rating.
> >
> > As for **biological inspiration**, the response is okay yet still provides too limited insightful analogy between the op and the bio to convince the readers that this is not a decoration of storytelling.
> >
> > As for **parameter efficiency**, the response is good.
> >
> > In short, I sincerely appreciate the authors' efforts in addressing my concerns and truly expect to see the authors providing the necessary results to make their novel work solid enough.

---

> > > ### Author Response · Authors · 2025-08-06
> > >
> > > We thank the reviewer for his reply and the acknowledgement of the Significance and Originality of our work.
> > >
> > > * **Primary CV tasks**: We agree, the dataset we used for testing segmentation is easier than the standard benchmark. This is why we presented them as preliminary results. However, we respectfully disagree that testing the mentioned primary CV task is necessary to evaluate our contribution. Our manuscript focuses on how higher-order correlations should be taken into account to fully grasp the content of a natural image. For this, we develop HoConv and show that its higher performance comes from it capturing those correlations. We delved into the analyses of sections 3.1 (synthetic textures) and 4 (Image statistics) to understand the mechanism behind its high performance. Although showcasing that HoConv could help in many CV tasks would be of interest, we believe that such a performance evaluation would be arid without understanding where that performance comes from.
> > >
> > > The reviewer’s pruning analogy seems to suggest our preliminary results might be trivial and meaningless. We want to emphasize that our work is fundamentally different from pruning studies—we're introducing a novel operator that captures higher-order statistics, not simply making existing architectures more efficient.
> > >
> > > * **Biological inspiration**: On the same line, it has been proved that biological vision takes into account those higher-order correlations already in the retina, that is, from the very first layer of the visual system. Accordingly, our HoConv improves classification performance by capturing those correlations as a front-end within the network, contrary to what deep CNNs do. Following the famous three levels of David Marr, the analogy is at the algorithmic level.
> > >
> > > * **Parameter efficiency**: Thanks for acknowledging it.

---

### Official Review · Reviewer_CXmf · 2025-07-01

**Clarity:** 3
**Significance:** 3
**Originality:** 3
**Rating:** 5
**Confidence:** 4

**Summary:**

The authors present an architecture named HoConv that extend standard CNNs with higher-order convolutions, and tested it on image classification.
A 2D input patch is first linearized into a vector x and then used to perform a set of convolution operations: 1D, 2D, 3D, etc..
Each time the input is the respective outer products (x, x * x, x * x * x, ...).
This helps capture some non-linear interactions between the features in x, not directly possible with one standard convolutional layer.

**Questions:**

What does the log ratio in Figure 4E show about HoConv? I am not sure how to interpret the visual patterns in it.

Did you consider applying your work on tasks beyond Image Classification? Some tasks might benefit greatly from the ability of HoConv to better capture highy-order image statistics.

Did you consider using a weights sum of the individual convolutions (1st order + w2 * 2nd order + w3 * 3rd order) with learnable weights instead of plain summation? I understand that w2 is just a scaling factor for the W2 matrix, but it would likely improve convergence since it is hard for the individual weights in W2 to learn the same scaling factor all at once.

**Ethical Concerns:**

["NO or VERY MINOR ethics concerns only"]

**Final Justification:**

The contribution is solid,  and the benefits of the proposed architecture are backed by strong experimental results.

**Limitations:**

Discussed in Sec. 5.1 to a good extent.

**Quality:**

2

**Strengths And Weaknesses:**

Strength:
- AFAIS, the proposed architecture is novel. Similar extensions to CNNs I am aware of are not based on outer products.
- The authors highlight the presence of tied weights as in inherent shortcoming of standard CNNs that HoConv helps mitigate.
- The authors designed experiments that demonstrate how HoConv is more sensitive to higher-order image statistics than standard CNNs, based on a deliberately generated texture dataset.

Weaknesses:
- The comparison with baseline CNNs in terms of accuracy is problematic. It is hard to maintain an exact number of parameters between both models.  Even then, those are not utilized in similar ways, and the FLOPS differences are huge. This latter issue alone gives an unfair advantage to the new method. To appreciate how important that is, consider re-training some popular ImageNet methods for a few more epochs. It is really easy to squeeze  ~1% accuracy improvement on ImageNet purely based on that.
- Following up on point 1, there are other ways to show the merits of a new architecture, beyond beating baselines on standard benchmarks. I really appreciate your experiment on texture classification which does that very well. I wish there were more targeted examples that highlight those merits. Wouldn't fine-grained semantic segmentation tasks be great candidates? A shallow U-Net struggles a lot with small objects (e.g. traffic lights in CityScapes) and with producing a smooth borders between objects. I felt these are excellent examples where better leveraging natural image statistics would give you an edge.
- AFAIS, the manuscript does not provide evidence that your method leverages natural image statistics (a key motivation for HoConv), such as the one cited from Olshausen and Field. Synthetic texture images were great to demonstrate higher-order image statistics, but those are far from natural images. The book by Hyvärinen et al provide a variety of further examples (building on the work of Olshausen, Field, and a variety of other authors):

Hyvärinen, Aapo, Jarmo Hurri, and Patrick O. Hoyer. Natural image statistics: A probabilistic approach to early computational vision. Vol. 39. Springer Science & Business Media, 2009.

Typos:
- hellinger -> capitalize
- on Texture classification -> de-capitalize

---

> ### Author Rebuttal · Authors · 2025-07-31
>
> Dear Reviewer CXmf,
>
> Thank you for your thoughtful review and valuable suggestions. We address your concerns below.
>
> ### W1: Fair comparison with baseline CNNs
>
> 1. You raise an important point about the FLOP differences: matching parameter numbers might not be fair if FLOPs are significantly different. To account for this, in App. A.1.2, LL464-472 we indeed compared HoCNN with a deep CNN having similar computational budget on CIFAR-10:
>
>    * Deep CNN: 7.09M FLOPs, 71.20% accuracy
>    * HoCNN: 6.81M FLOPs, 72.06% accuracy
>
>    Even with slightly fewer FLOPs, HoCNN achieves 1.21% higher accuracy, demonstrating that improvements aren't merely from increased computation.
>
> 2. **Parameter-matched comparisons**: Throughout our experiments (Table 2), we ensured HoCNN has comparable or fewer parameters than baseline CNNs while achieving consistent improvements. In Appendix A.1 (second paragraph) we also report a case where HoConv outperforms a CNN with 25% more parameters.
>
>
> ---
> ### W2: Applications beyond standard benchmarks
>
> We greatly appreciate your insightful suggestion about exploring tasks beyond classification. Your comment motivated us to conduct additional experiments, and we're pleased to report promising initial results.
> Following your suggestion, we investigated fine-grained visual tasks using the CUB-200-2011 dataset (Wah et al 2011). This dataset, consisting of 200 bird species (classes) is one of the benchmarks of fine grained classification and also presents segmentation masks, it  is particularly challenging due to subtle inter-class differences and limited training data (~6k images), making it an excellent test case for our approach.
>
> **New results:**
>
> 1. **Fine-grained classification**: We fine-tuned ImageNet-pretrained ResNet18 and HoResNet18 on CUB-200-2011. Our HoResNet18 showed even larger improvements than on standard benchmarks (72.31 % +- 0.29% vs ResNet18’s 68.24 % +- 0.37%), suggesting higher-order convolutions are particularly beneficial for capturing subtle discriminative patterns. Our result is inline with bilinear pooling CNNs (cite) which are somehow similar to our HoConv, even though the "higher-order" mechanism is at the level of the fully connected (last) layer in those
>
> 2. **Segmentation**: A key challenge was developing a transposed version of our higher-order convolution for decoder architectures. We successfully implemented this and built U-Net architectures using our pretrained encoders. The HoResNet18-based U-Net consistently outperformed the ResNet18 baseline in mean IoU (87.72 % +- 0.68% mIoU vs ResNet18’s 82.17 % +-  0.75%), demonstrating effective transfer to dense prediction tasks.
>
> These results validate your intuition that tasks requiring precise local pattern detection benefit from our approach, we will include them in the paper, and we are extremely grateful to this Reviewer for prompting us to explore this novel direction: we will add these results to the appendix. However, even if these preliminary results are very encouraging, we believe that an in depth analysis on other datasets and models would be outside the scope of our paper. Here, instead of testing multiple computer vision tasks, we preferred to focus on image classification, and deepen the understanding of why/how our HoConv layer is effective in capturing higher order correlations in natural images.
>
> ---
> ### W3: Connection to natural image statistics
>
> We acknowledge that our synthetic textures differ from natural images. However,  our perturbation analysis (Section 4 and Fig. 4A and B) exactly show that HoConv leverages higher order correlations to perform classification. There we demonstrate that HoCNN's increased sensitivity to higher-order statistical perturbations in natural images (CIFAR-10) translates to better classification performance, suggesting the model leverages these statistics effectively.
>
> Additionally,  **Koenderink (2018) alignment**, our finding that performance peaks at 3rd/4th order directly aligns with natural image statistics where quadratic, cubic, and quartic powers dominate approximately 63%, 35%, and 2% of pixels respectively (Section 5).
>
> Finally, the benchmark models we used for comparison have been developed to perform classification tasks on those datasets, so we felt that we should outperform them within the domain they were developed for to make it a fair comparison. If we select an arbitrary database, showing that our architecture outperforms others may raise suspicion that this result reflects our choice of database, rather than the strength of our model.
>
>
> ---
> ### Q1: Interpretation of log ratio in Figure 5E
> The log ratio visualization (Figure 5E) highlights regions where the representational geometries of CNN and HoCNN differ most. Positive values (red) indicate image pairs that HoCNN represents as more dissimilar than CNN, while negative values (blue) show the opposite. The structured pattern reveals that HoCNN creates more distinct class-specific representations, particularly enhancing inter-class dissimilarity while maintaining intra-class similarity. This is complemented by Figure 11 in Appendix where it is possible to appreciate the increased dissimilarity in HoConv’s representations from a distribution perspective.
>
> ---
> ### Q2: Applications beyond image classification
> Your question prompted valuable exploration. Beyond the CUB-200-2011(Wah et al 2011) results mentioned above, we see potential for higher-order convolutions in:
>
> - **Medical imaging**: Where subtle texture patterns and local correlations are diagnostic
>
> - **Dense Prediction Tasks**: Our new transposed higher-order convolution enables full encoder-decoder architectures for segmentation
>
> The development of transposed higher-order convolutions significantly expands the applicability of our approach beyond classification. In future work, we intend to explore these directions in more detail.
>
> ---
> ### Q3: Weighted summation of orders
> This is an excellent suggestion. We actually implemented and tested this approach during our initial experiments, using learnable coefficients to weight different orders (1st + w2×2nd order + w3×3rd order). Interestingly, we didn't observe consistent improvements over our current implementation with plain summation and normalization factor $s = 1/\sqrt{n_V}$ (Section 2).
> The learned weights tended to converge to similar values across runs, suggesting our normalization strategy already provides appropriate scaling. However, we agree that explicit learnable weights could offer more flexibility for certain applications and merit further investigation in domain-specific contexts.
>
> ---
> ### Typo corrections
> Thank you for catching these:
> - We'll capitalize "Hellinger" throughout
> - We'll change to "texture classification" (lowercase)
>
> We sincerely thank you for your stimulating comments that led to these additional insights and results. Your suggestions have significantly strengthened our work and we hope that our response helped to clarify your doubts.

---

> > ### Comment · Reviewer_CXmf · 2025-08-04
> >
> > I appreciate the response by the authors and the experiments they conducted to address the points I raised.
> > I am raising my initial review score accordingly.

---

### Official Review · Reviewer_tv7t · 2025-07-02

**Clarity:** 2
**Significance:** 2
**Originality:** 2
**Rating:** 4
**Confidence:** 4

**Summary:**

The article aims to build biologically inspired models for computer vision, based on recently proposed Volterra neural networks. Higher-order CNNs are proposed to achieve this goal, by extending existing CNNs architectures. Through standard benchmarks and synthetic datasets, high-order CNNs are shown to outperform traditional CNNs. Perturbation analysis and representational similarity analysis further reveal that different order of convolution process distinct statistical properties of natural images.

**Questions:**

-	Clarify how the PCA is done in Section 2.1, i.e. on what vectors?
-	In general, traditionally CNNs can also capture well 2nd order information in natural images, in particular for texture classification since CNNs models can lead to performance improvement (see e.g. Deep Filter Banks for texture recognition and segmentation 2010). I am thus not convinced by the results in Table 1 and Fig 2 which shows that the baseline CN performs poorly. This point should be justified under some constraints of baseline CNNs, such as the depth / width/ filter size limitation. It seems not so clear how the baseline CNN is chosen. This makes the methodology part of the results in this article less convincing.
-	On line 193, where in Appendix A.1 you performed additional experiments with a deep and more complex CNN?
-	What is the size of L and order in HoResNet-18 model on line 204? How about its computational speed compared to the other models compared in Table 2?
-	The ImageNet result in Table 2 has no +/- analysis. At least 5-10 simulations should be made to estimate this number to show that the improvement of HoResNet-18 is significant.

**Ethical Concerns:**

["NO or VERY MINOR ethics concerns only"]

**Final Justification:**

I thank the authors for their responses which have addressed my concerns. I am happy to raise my score to accept the article given their original contributions to develop more advanced CNNs for classification problems.

I think the choice of the distance in the RSA analysis in Section 4 should be further discussed (i.e. how to Construct Representational Dissimilarity Matrices, based on e.g. correlation distance, Euclidean distance or the Mahalanobis distance). This may give different conclusions, as suggested in [Representational similarity analysis – connecting the branches of systems neuroscience, 2008].

**Limitations:**

yes

**Quality:**

2

**Strengths And Weaknesses:**

Strength: The article makes a step forward to build higher-order CNNs, inspired by Volterra neural networks. Contrary to the latter approach, which is built upon polynomial functions, the proposed model is built upon a cascade of polynomial functions and standard ReLU non-linear CNNs layers. A careful quantitative analysis on the benefit of this approach is done on synthetic texture datasets, as well as image classification benchmarks.

Weakness: Certain points on the weakness of CNNs should be made in a more careful way. This makes the results on the role of higher convolution in CNNs less convincing. Numerical results on ImageNet should also be more conclusive. A better explanation of RSA in Figure 4 would be helpful to understand the results.

---

> ### Author Rebuttal · Authors · 2025-07-31
>
> Dear Reviewer tv7t,
>
> Thank you for your review and constructive feedback. We address your concerns below.
>
> ### Q1: PCA methodology clarification (Section 2.1, L 130-135)
>
> The PCA analysis is performed on the activations after the convolutional block (Conv/HoConv + BatchNorm + ReLU + Max Pooling). Specifically:
> - We generate a fixed - white noise - synthetic input image containing binary textures with all possible 1-, 2-, 3-, and 4-point gliders.
> - We randomly initialize the network 10,000 times
> - For each initialization, we extract the flattened activation vectors after the first convolutional block
> - PCA is performed on these activation vectors (shape: 10,000 × activation_dimension)
>
> This methodology isolates the architectural inductive biases by examining how different weight initializations affect the representation space. We apologize if our explanations were not exhaustive. We will add these clarifications to the camera-ready manuscript.
>
> ----
> ### Q2: CNN baseline performance on texture classification
>
> We appreciate this important critique. You are correct that standard CNNs can capture second-order statistics effectively in many contexts, especially with **sufficient depth**. Our baseline CNN's poor performance on the synthetic texture dataset (Table 1) is indeed due to specific architectural constraints:
>
> 1. **Kernel size limitation**: We used 2×2 kernels motivated by the texture generation process (gliders are defined on 2×2 patches) in the first layer;
> 2. **Shallow architecture**: Only 2 convolutional layers to isolate the effect of higher-order operations
> 3. **Limited capacity**: 10 kernels in first layer, 2 in second layer
>
> We have chosen these settings to perform a controlled comparison between CNN and HoConv, not to push their performance at maximum. Indeed, as shown in Appendix A.2, larger kernels (5×5, 7×7) in standard CNNs improve image classification - and so they would do for our HoConv. Yet, this also suggests that while larger receptive fields can help capturing higher-order correlations, our explicit higher-order formulation provides a more generalizable solution even for small kernels. We will explain this clearly in the camera-ready manuscript
>
> ----
> ### Q3:App. A.1,  Location of deep CNN experiments
>
> It’s in the second paragraph of A.1. We apologize for the confusion due to an unclear organization of appendix A.1. We will introduce a subsection “Comparison with deeper CNNs”. The deep CNN architecture has:
> - Eight convolutional blocks (3×3 kernels)
> - Channel progression: 8→8→16→16→32→32→64→64
> - Total parameters: 108,130 (25% more than HoCNN)
> - Performance: 71.20% on CIFAR-10 (vs 72.35% for HoCNN)
>
> ----
> ### Q4: HoResNet-18 architecture details
>
> In HoResNet-18:
> - **Order**: We use up to 3rd order expansions (1st + 2nd + 3rd order terms)
> - **Location**: Higher-order blocks are used only in Stage 1 (first 2 residual blocks)
> - **Channels**: 30 channels in the higher-order stage (vs 64 in standard ResNet-18)
>
> As already reported in App. A.1.1, the computational speed values are:
> - Standard ResNet-18: 1.82 GFLOPs
> - HoResNet-18: 3.00 GFLOPs (1.65× increase)
>
> ----
> ### Q5: Statistical significance for ImageNet
>
> You raise a valid point about the lack of error analysis for ImageNet results. Due to computational constraints, we performed single runs at the time of the submission. However, we acknowledge this limitation and agree that multiple runs would strengthen our claims. We have been able to collect results for 3 runs at the moment (we are currently running more to get more reliable statistics), results are reported in the table below:
>
>
> | Run  | ResNet-18 (%)   | HoResNet-18 (%)  | Difference (%) |
> |------|-----------------|------------------|----------------|
> | 1    | 68.53           | 69.38            | +0.85          |
> | 2    | 68.72           | 69.61            | +0.89          |
> | 3    | 68.60           | 69.40            | +0.82          |
> | Mean | 68.62 ± 0.10    | 69.46 ± 0.13     | +0.84          |
>
>
> ----
> ### Additional clarifications on RSA (Figure 5)
>
> Regarding the RSA analysis, we compare representational geometries between CNN and HoCNN:
> - We extract activations from 100 CIFAR-10 test images (10 per class, randomly selected)
> - Compute pairwise distances between activation vectors
> - Construct Representational Dissimilarity Matrices (RDMs)
> - The log-ratio visualization (panel E) highlights where representations differ most
>
> The key finding is that HoCNN creates more dispersed representations (Larger Log-ratios in Fig. 4E and Figure 11D of Appendix A.9.1), enabling better class separation. We will increase manuscript clarity here.
>
> We hope these clarifications address your concerns and demonstrate the rigor of our approach.

---

> ### Author Response · Authors · 2025-08-05
>
> Dear Reviewer tv7t,
>
> Following the NeurIPS organizers' encouragement to continue discussions, we wanted to follow up on our previous response and provide additional results that may help address your concerns.
>
> Regarding statistical significance for ImageNet (Q5), we've now completed runs with 5 different random seeds. The results consistently show improvement with HoResNet-18:
>
> | Run  | ResNet-18 (%)   | HoResNet-18 (%)  | Difference (%) |
> |------|-----------------|------------------|----------------|
> | 1    | 68.53           | 69.38            | +0.85          |
> | 2    | 68.72           | 69.61            | +0.89          |
> | 3    | 68.60           | 69.40            | +0.82          |
> | 4    | 68.67           | 69.52            | +0.85          |
> | 5    | 68.58           | 69.45            | +0.87          |
> | Mean | 68.62 ± 0.08    | 69.47 ± 0.09     | +0.85 ± 0.03   |
>
>
> These additional runs confirm our earlier findings with reduced variance, showing a consistent and statistically significant improvement of ~0.85% across all seeds.
>
> Have our responses and this additional experimental result helped clarify your concerns? We're particularly interested in your thoughts on whether these improvements, combined with our analysis of how HoConv captures higher-order statistics, demonstrate sufficient merit for our approach.
>
> We appreciate your time and valuable feedback throughout this process.

---

> ### Comment · Area_Chair_np9c · 2025-08-07
> **Please engage with the rebuttal ASAP**
>
> Dear reviewer,
>
> Thank you for reviewing this paper. The authors have provided a detailed rebuttal to your review, but I cannot see any edits to your review that take the author reply into account, and also your "Final justification" is missing. **Please engage with to the rebuttal asap**. Note I flagged you as non-responsive, but this flag will be removed once you posted a satisfactory reply
>
> Thank you, the AC

---

### Official Review · Reviewer_qSKJ · 2025-07-05

**Clarity:** 3
**Significance:** 4
**Originality:** 3
**Rating:** 5
**Confidence:** 4

**Summary:**

This paper introduces Higher-Order Convolutions (HoConv), a learnable extension of classical linear convolution operators that captures higher-order correlations in natural images beyond first-order (linear) relationships. Inspired by the observation that natural scenes exhibit complex statistical dependencies, the authors design a convolutional block that computes and sums responses from filters of order 1 through 4. Each order captures different correlation structures in the image, and the design ensures parameter independence across orders to facilitate more flexible learning. Compared to classical CNNs of similar or even higher parameter counts, HoCNNs consistently outperform on multiple image classification benchmarks. The authors also provide insight into how these higher order convolutions work, by the representational capacity of HoCNNs via synthetic data and PCA-based visualizations.

**Questions:**

1. Synthetic image visualization: Could you include the binary texture image referenced in Figure 1B and Section 2.1, or at least a representative example? This would help readers better understand what it means to include “all 1–4 point correlations” in 2×2 patches.
2. PCA experiment design: In your PCA comparison of CNN and HoCNN representations, each point corresponds to a randomly initialized model. Can you clarify the rationale for this setup? Wouldn't it be more informative to fix one model and vary the input instead?
4. Typo in Figure 4: Please clarify the inconsistency between the image panels and the figure caption in Figure 4. Is panel C the HoConv block or the Conv block?
5. Extension to SOTA models: You mention potential connections with transformer-based vision models. Can you speculate on how higher-order convolutions could be integrated into hybrid or attention-based architectures?

**Ethical Concerns:**

["NO or VERY MINOR ethics concerns only"]

**Final Justification:**

This is a technically solid paper and the authors have adequately addressed my remaining questions and concerns in their rebuttal. Therefore, I believe it should be accepted, and I am maintaining my score of 5.

**Limitations:**

Yes

**Quality:**

4

**Strengths And Weaknesses:**

### Strengths

- Clear motivation and originality: The paper is well-motivated by limitations of traditional CNNs in capturing higher-order pixel correlations. The proposed approach is a principled extension of convolution, grounded in signal statistics and biological plausibility.
- Novel architecture: While prior work has explored 2nd-order filters, this paper generalizes the idea up to 4th order in a clean, modular way, while ensuring learnability and independence of parameters across orders.
- Strong experimental design: The authors test the model across a variety of vision benchmarks and provide statistical evidence (via repeated experimental runs) to support the robustness of their improvements.
- Analysis beyond benchmarks: The paper goes beyond classification accuracy to examine what different orders of convolution capture, using PCA and synthetic datasets.
- Clarity: The manuscript is overall very well-written and accessible. The motivation, implementation details, and experiments are communicated clearly.

### Weaknesses

- Visualization gap: In Figure 1B and section 2.1, the authors reference a synthetic image containing binary textures that exhibit all 1–4 point correlations within 2×2 patches. This image is not shown, and it’s difficult to develop intuition for it without a visual example.
- Questionable analysis choice: In the PCA-based analysis comparing CNN vs HoCNN representations, each data point corresponds to a random weight initialization, rather than a fixed network processing different images. This design decision may limit the interpretability of the representation space analysis.
- Minor clarity/consistency issues: Figure 4 panel labels (C and D) are mismatched between figure and caption. Similarly, textual references to these panels are inconsistent.

---

> ### Author Rebuttal · Authors · 2025-07-31
>
> Dear Reviewer qSKJ,
>
> Thank you for your thorough and constructive review. We appreciate your positive assessment of our work and your recognition of its contributions to understanding higher-order visual processing. We address your specific concerns and questions below.
>
> ----
> ### W1 & Q1: Visualization of synthetic binary textures
> We acknowledge this omission. The synthetic image referenced in Figure 1B and Section 2.1 contains randomly distributed 2×2 binary patches designed to exhibit all possible gliders. Specifically what we mean by “ including “all 1–4 point correlations” in 2×2 patches” is that if you look at the distribution of all the possible 2x2 patches, you’ll see representatives of the 1, 2, 3, 4 point (i.e. pixel) gliders.
> As an intuition you can think about binary (either black or white pixels) white noise: since we’re not imposing any constraint  on neighbouring pixels, in each realization there will be a number of 1, 2, 3 and 4 point (i.e. pixel) correlations.
>  While we cannot add figures in the rebuttal, we will include a comprehensive visualization panel in the revised manuscript showing it.
>
> ----
> ### W2 & Q2: PCA experiment design rationale
> We appreciate your insight regarding this analysis choice. Our decision to use multiple random initializations and a fixed input image rather than multiple input images and a single random initialization was motivated by the goal of understanding the intrinsic representational capacity of our architecture without training. More in detail: since we are not training our model and the baseline CNN in this experiment, the models haven’t learned how to separate different N-point correlations yet. In this way we only wanted to highlight the fact that the HoCNN has potentially more representational power: more PCs are needed which means higher dimensionality of the representation.
>
> While we used this setting for this experiment, we’d like to highlight that for Representational Similarity Analysis in the last section of the paper we are comparing multiple images (with trained networks) from different classes, and representational differences are still present and leading to a gap in performance.
>
> ----
> ### W3 & Q3: Figure 4 inconsistency
> Thank you for catching this error. The correct labeling is:
> Panel C: Higher-order convolution (HoConv) block
> Panel D: Standard convolution (Conv) block
> We will correct all references in the text and ensure consistency throughout the manuscript.
>
> ----
> ### Q4: Relationship with Transformers and Self-Attention
> Thank you for this question, it touches upon a crucial point regarding the relationship between our proposed higher-order convolutions and the attention mechanisms that empower modern SOTA architectures like Vision Transformers (ViTs). While both approaches are designed to capture complex feature interactions, they do so through mathematically distinct and complementary mechanisms.
>
> To answer, we can start with the math behind HoCNNs and ViTs:
>
> Our Higher-Order Convolution (HoConv) extends the standard convolution. For a given input patch $\mathbf{x}$, the output $y(\mathbf{x})$ up to the 3rd order is:
> $y(\mathbf{x}) = \sum_i w_1^i x_i+ \sum_{i,j} w_2^{ij} x_i x_j + \sum_{i,j,k} w_3^{ijk} x_i x_j x_k + \dots$
>
> The Scaled Dot-Product Attention used in Transformers is formulated as:
> $\text{Attention}(Q, K, V) = \text{softmax}\left(\frac{QK^T}{\sqrt{d_k}}\right)V$
>
> Here, the Query ($Q$), Key ($K$), and Value ($V$) matrices are linear projections of the input tokens $\mathbf{X}$ (e.g., $Q = \mathbf{X}W_Q$, $K = \mathbf{X}W_K$, $V = \mathbf{X}W_V$).
>
> **Parallels and Differences**
> 1. The 2nd-Order Parallelism:
> The most direct parallel exists between our 2nd-order term and the core of the self-attention mechanism. The term $QK^T$ in attention can be expanded as:
> $QK^T = (\mathbf{X}W_Q)(\mathbf{X}W_K)^T = \mathbf{X}(W_Q W_K^T)\mathbf{X}^T$
> This is a quadratic form of the input features $\mathbf{X}$. It computes multiplicative interactions between all pairs of input tokens. Similarly, our 2nd-order convolution, $\sum w_2^{ij} x_i x_j$, is also a quadratic form, but computed over a local patch.
> Parallelism: Both mechanisms model feature relationships through pairwise multiplicative interactions (a quadratic operation).
>
> 2. The 3rd-Order Disparity:
> One might be tempted to see a 3rd-order interaction in attention through the final multiplication by $V$. The full attention operation is approximately $\text{softmax}(\mathbf{X}\mathbf{X}^T)\mathbf{X}$. However, this is not a true 3rd-order polynomial. The softmax function is a significant nonlinearity that normalizes the quadratic interactions to form weights. The subsequent multiplication by $V$ is a re-weighting or value-mixing operation, not a simple cubic interaction.
>
> **Difference**: Our 3rd-order convolution is a direct, learnable cubic polynomial ($\sum w_3^{ijk} x_i x_j x_k$). In contrast, attention's three-term structure computes dynamic weights based on pairwise similarity and applies them to a value representation. It does not explicitly model learnable 3-point multiplicative interactions in the same way.
>
> In conclusion, HoConv provides a powerful mechanism for learning complex local feature detectors with a strong spatial inductive bias. Self-attention provides a flexible mechanism for learning global, long-range dependencies with dynamic, content-based routing of information.
>
> These differences suggest promising avenues for future research:
>
> * Hybrid architectures: Integrating HoConv in early layers of transformer networks could enhance local feature extraction before global attention processing.
>
>
> * Order-selective attention: Using attention mechanisms to selectively emphasize different orders of convolution based on image content.
>
>
> Our work on higher-order convolutions therefore offers not only immediate performance improvements but also opens new research directions for integrating complementary approaches to visual representation learning. We have included a brief discussion of these potential extensions in the revised Discussion section, and we thank the Reviewer for prompting us to include these interesting suggestions that may spur further developments in the field.

---

> > ### Comment · Reviewer_qSKJ · 2025-08-07
> > **Thanks for your thoughtful response**
> >
> > The authors have addressed all of my remaining questions and concerns. I am maintaining my score to accept (5).

---

> ### Author Response · Authors · 2025-08-05
>
> Dear Reviewer qSKJ,
>
> Following the NeurIPS organizers' encouragement to continue discussions, we wanted to follow up on our rebuttal to see if you had any additional questions or if there were aspects of our response that could benefit from further clarification.
> We particularly appreciate your positive assessment of our work and your thoughtful questions about the synthetic image visualization, PCA experiment design, figure inconsistencies, and potential extensions to SOTA models. In our rebuttal, we addressed these points in detail, including our plans to:
>
> * Add comprehensive visualization panels of the binary textures in the revised manuscript
> * Clarify our rationale for the PCA experiment design focusing on intrinsic representational capacity
> * Correct the labeling inconsistency in Figure 4
> * Expand on the relationship between our HoConv approach and transformer-based architectures
>
> We hope that our responses have helped you to clarify your concerns.
>
> Thank you for your time and valuable feedback throughout this process.

---

### Comment · Area_Chair_np9c · 2025-08-04
**Please engage with the rebuttals**

Dear reviewers,

The authors have provided detailed rebuttals. If you haven't done so, please engage with the rebuttal _by updating your review_ and by _providing a final justification_, asking for further clarification where needed.

Thank you, the AC

---

### Decision · Program_Chairs · 2025-09-17

**Decision:**

Accept (poster)

**Comment:**

This paper presents a compelling and novel extension to CNNs through the introduction of learnable, biologically-inspired higher-order convolutions (HoConv), which employ a Volterra-like expansion to capture multiplicative interactions and higher-order statistical dependencies in natural images. This work stands out through its clear biological motivation as well as its careful experimental design and analysis, which demonstrates consistent performance gains over CNN baselines across multiple benchmarks and provides valuable analysis of the distinct representational properties of each convolution order. While reviewers tv7t and CXmf initially raised valid concerns regarding the choice of baselines, the statistical significance of the ImageNet results, the limited task validation beyond classification, and the depth of the biological inspiration, the authors' rebuttal effectively addressed the majority of these points and they increased their score. Although Mz2V maintained that validation on primary vision tasks like detection and segmentation on modern benchmarks (e.g., MS COCO) would be ideal, the AC finds the paper's contributions to be sufficiently significant, technically solid and insightful for acceptance in its current form.